# SecureGS: Boosting the Security and Fidelity of 3D Gaussian Splatting Steganography

**Xuanyu Zhang**[1*], **Jiarui Meng**[1*], **Zhipei Xu**[1], **Shuzhou Yang**[1], **Yanmin Wu**[1],
**Ronggang Wang**[1,2], **Jian Zhang**[1,2†]
[1]School of Electronic and Computer Engineering, Peking University
[2]Guangdong Provincial Key Laboratory of Ultra High Definition Immersive Media Technology,
Shenzhen Graduate School, Peking University

## Abstract

3D Gaussian Splatting (3DGS) has emerged as a premier method for 3D representation due to its real-time rendering and high-quality outputs, underscoring the critical need to protect the privacy of 3D assets. Traditional NeRF steganography methods fail to address the explicit nature of 3DGS since its point cloud files are publicly accessible. Existing GS steganography solutions mitigate some issues but still struggle with reduced rendering fidelity, increased computational demands, and security flaws, especially in the security of the geometric structure of the visualized point cloud. To address these demands, we propose a **SecureGS**, a secure and efficient 3DGS steganography framework inspired by Scaffold-GS's anchor point design and neural decoding. SecureGS uses a hybrid decoupled Gaussian encryption mechanism to embed offsets, scales, rotations, and RGB attributes of the hidden 3D Gaussian points in anchor point features, retrievable only by authorized users through privacy-preserving neural networks. To further enhance security, we propose a density region-aware anchor growing and pruning strategy that adaptively locates optimal hiding regions without exposing hidden information. Extensive experiments show that SecureGS significantly surpasses existing GS steganography methods in rendering fidelity, speed, and security.

## 1 Introduction

Benefiting from its real-time rendering capabilities and impressive rendering quality, 3DGS has become a mainstream 3D representation approach. Since optimizing a 3D scene requires a large amount of computing resources and 3D tampering approaches are developing rapidly, protecting the copyright and privacy of 3D assets is particularly important. As an emerging research field, 3DGS steganography aims to embed bits, images, or 3D content into 3D Gaussian points invisibly, and extract them losslessly in the decoding end. It has great potential for application in encrypted communications, copyright protection, identity verification, and forensic analysis.

As a preliminary work for 3DGS steganography, previous works on NeRF watermarking have achieved excellent results. For instance, StegaNeRF (Li et al., 2023a) jointly finetuned the weights of NeRF and a decoder so that each 2D rendered view can decode the pre-defined watermark. CopyRNeRF (Luo et al., 2023b) and WateRF (Jang et al., 2024a) focused on bit hiding and robust decryption and can achieve copyright traceability of NeRF via a 2D view. However, these methods cannot be effectively applied to 3DGS steganography, since 3DGS is an explicit representation and its point cloud files are often made public and transparent for online real-time rendering.

To achieve this demand, GS-Hider (Zhang et al., 2024b) used a coupled feature field and neural decoders to simultaneously render the original and hidden scene, as shown in Fig. 1(a). However, it presents several shortcomings in terms of fidelity, security, and rendering speed. **Fidelity:** Representing the original scene and hidden information using the same set of Gaussian points with a compact

---
* Equal contributions, † Corresponding author. This work was supported in part by Guangdong Provincial Key Laboratory of Ultra High Definition Immersive Media Technology (No. 2024B1212010006) and Shenzhen General Research Project (No. JCYJ20241202125904007).

Figure 1: Analysis of previous 3DGS steganography method GS-Hider (Zhang et al., 2024b).

feature attribute will easily lead to mutual interference, especially when the geometry of the hidden information is inconsistent with that of the original scene, resulting in suboptimal rendering fidelity. **Rendering Speed:** Due to the use of convolutional networks to decode the rendered coupled feature fields, there is an increase in computational complexity, which affects the rendering speed of 3DGS steganography to a certain extent. **Security:** We categorize the security of GS steganography into two types, namely file format security and geometric structure security. Format security means that the published 3DGS point cloud does not add any additional attributes that could arouse suspicion or lead to deletion by malicious abusers. Geometric structure security denotes when visualizing the Gaussian point cloud, no traces of the hidden scene's geometric structure are revealed. The latter is what GS-Hider cannot achieve, which may pose a risk of leaking hidden messages (such as Fig. 1(b)).

Our insight for solving these problems comes from a successful variant of 3DGS, namely Scaffold-GS (Lu et al., 2023). Scaffold-GS uses anchor points to establish a hierarchical 3D scene representation. Then, a set of neural Gaussians with learnable attributes are dynamically predicted from the anchor feature via several simple MLPs. Scaffold-GS is naturally suited for steganography for two reasons: **First**, it uses implicit MLPs to store Gaussian point attributes, allowing encryption to focus on MLPs rather than explicit Gaussian points. **Second**, it maintains comparable rendering quality and real-time performance to 3DGS, preserving its high fidelity and fast rendering benefits.

To address the above limitations, we propose a more secure and efficient GS steganography framework, dubbed **SecureGS**. Fig. 2 presents our application scenario. Specifically, we design a hybrid decoupled Gaussian encryption representation capable of decoding two sets of dense Gaussian points from a sparse set of anchor points, which are used for rendering the original scene and the hidden object, respectively. Note that we specifically introduce an offset predictor to conceal the positions of the hidden Gaussian points. Furthermore, to prevent the geometric structure of the hidden object from being exposed, we innovatively propose a density region-aware anchor growing strategy. Based on the gradient of the joint rendering loss, it can adaptively find the location of hidden 3D objects, thereby lowering the splitting threshold in that region and allowing the original scene's anchors to cover the hidden anchor points safely. In a nutshell, our contributions are summarized as follows.

❑ (1) We present a novel attempt to introduce neural voxelization into the 3DGS steganography, realizing a more secure, high-fidelity, and efficient framework **SecureGS**. It can effectively hide 3D objects, images, and bits within the original 3D scene and precisely extract them.

❑ (2) We develop a hybrid decoupled Gaussian encryption representation that can hide and predict the location and attributes of the hidden Gaussian points via a set of neural decoders, ensuring that the overall framework maintains a format consistency in the point cloud files and is efficient in rendering hidden message and preserving the original scene.

❑ (3) We propose a region-aware density optimization that can locate hidden 3D positions and promote anchor growth in that region while inhibiting pruning, thereby significantly enhancing the geometric structure security of the point cloud within the overall framework.

❑ (4) Extensive experiments demonstrate that our method significantly outperforms existing 3D steganography methods in terms of the fidelity of container 3DGS, rendering speed, and security.

## 2 RELATED WORKS

### 2.1 3D GAUSSIAN SPLATTING

3D Gaussian Splatting (3DGS) (Kerbl et al., 2023) has emerged as a highly effective approach for 3D scene reconstruction, utilizing millions of 3D Gaussians. Starting from a set of Structure-from-Motion

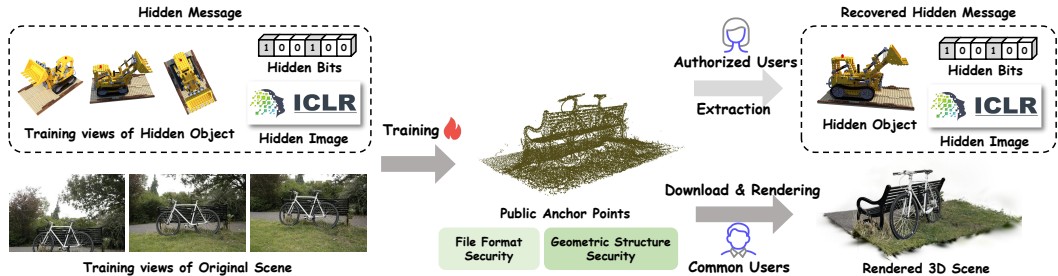

Figure 2: Overall Pipeline of our SecureGS. 3D objects, images, and bits can be hidden in the original 3D scene, and only authorized users can decode these hidden messages. The core of our method is to ensure both the file format and geometric structure security of the public anchor points.

(SfM) points, each point is assigned as the center (mean) $\boldsymbol{\mu}$ of a 3D Gaussian distribution:

$$\mathcal{G}(\mathbf{x}) = e^{-\frac{1}{2}(\mathbf{x}-\boldsymbol{\mu})^\top \boldsymbol{\Sigma}^{-1}(\mathbf{x}-\mu)}, \quad \boldsymbol{\Sigma} = \mathbf{R}\mathbf{S}\mathbf{S}^\top\mathbf{R}^\top, \tag{1}$$

where $\mathbf{x}$ is a position within the 3D scene and $\boldsymbol{\Sigma}$ denotes the covariance matrix of the 3D Gaussian, which is formulated using a scaling matrix $\mathbf{S}$ and rotation matrix $\mathbf{R}$. Instead of using the resource-intensive ray-marching, 3DGS efficiently renders the scene via a tile-based rasterizer. Specifically, the 3D Gaussian $\mathcal{G}(\mathbf{x})$ are first transformed to 2D Gaussians $\mathcal{G}'(\mathbf{x})$ on the image plane via the projection mechanism (Zwicker et al., 2001). Let $\mathbf{C} \in \mathbb{R}^{H \times W \times 3}$ represent the color of the rendered image where $H$ and $W$ represent the height and width of images, the $\alpha$-blending process of a tile-based rasterizer is outlined as follows:

$$\mathbf{C}[\mathbf{p}] = \sum_{i=1}^{N} \boldsymbol{c}_i \sigma_i \prod_{j=1}^{i-1}(1 - \sigma_j), \quad \sigma_i = \alpha_i \mathcal{G}'_i(\mathbf{p}), \tag{2}$$

where $\mathbf{p} = (u, v)$ is the queried pixel position and $N$ denotes the number of sorted 2D Gaussians associated with the queried pixel. $\boldsymbol{c}_i$ and $\alpha_i$ respectively denote the color and opacity component of the Gaussian point. In contrast to previous implicit representations, 3DGS dramatically improves both training speed and rendering efficiency. To further elevate 3DGS's rendering capabilities, Mip-Splatting (Yu et al., 2023b) introduced 2D and 3D filtering, enabling high-quality, alias-free rendering across arbitrary resolutions. Moreover, Scaffold-GS (Lu et al., 2023) integrated structured neural anchors to enhance the rendering quality from diverse perspectives. The exceptional performance of 3DGS has extended its utility to a broad range of applications, including SLAM (Keetha et al., 2023; Matsuki et al., 2023), 4D reconstruction (Li et al., 2023b; Luiten et al., 2023; Wu et al., 2023; Yang et al., 2024b), and 3D content generation (Tang et al., 2023; Yi et al., 2024; Yang et al., 2024a).

## 2.2 3D STEGANOGRAPHY

Steganography has undergone significant evolution over the decades (Provos & Honeyman, 2003; Cheddad et al., 2010). With the rise of deep learning, deep steganography methods have been developed to invisibly embed messages into various carriers and reliably extract them, spanning 2D images (Zhang et al., 2024a; Zhu et al., 2018; Baluja, 2019; Yu et al., 2023a; Xu et al., 2024; Zhang et al., 2024c), videos (Zhang et al., 2024d; Luo et al., 2023a), audio (Liu et al., 2023a;b; Chen et al., 2023; Roman et al., 2024), and generative models (Wen et al., 2024; Fernandez et al., 2023). Traditional 3D steganography has focused on watermarking explicit 3D representations, such as meshes (Ohbuchi et al., 2002; Praun et al., 1999; Wu et al., 2015), typically by perturbing vertices or transforming data into the frequency domain. Additionally, Yoo et al. (Yoo et al., 2022) proposed a method to extract copyright information from individual 2D views, even without the full 3D mesh.

Recently, NeRF watermarking has garnered increased attention (Luo et al., 2023b; Jang et al., 2024a; Li et al., 2023a). For instance, StegaNeRF (Li et al., 2023a) embedded images or audio within 3D scenes by fine-tuning NeRF's weights, while CopyRNeRF (Luo et al., 2023b) introduced a watermarked color representation and a distortion-resistant rendering strategy to ensure robust message extraction. WaterRF (Jang et al., 2024a) leveraged deferred backpropagation with patch loss and employed discrete wavelet transform to enhance fidelity and robustness. NeRFProtector (Song et al., 2024) utilized a watermarking base model and progressive global distillation to explore relations

between rendering strategies and watermark embedding. Targeted at 3DGS, GS-Hider (Zhang et al., 2024b) used coupled feature fields and neural decoders to render the original and hidden scene, achieving high-quality 3D scene hiding. Meanwhile, GaussianStego (Li et al., 2024) proposed a generalized pipeline for information hiding and recovery in the 3D generation model. 3D-GSW (Jang et al., 2024b) presented a frequency-guided Densification strategy for robust watermarking in 3DGS. However, these methods do not take into account the security of point clouds, and the fidelity of container 3DGS is unsatisfactory.

## 3 METHODOLOGY

### 3.1 ANALYSIS OF THE PREVIOUS 3DGS STEGANOGRAPHY METHOD

We first review the recent 3DGS steganography work GS-Hider. As shown in Fig. 1(a), GS-Hider follows a "coupling-decoupling" process. It replaces the spherical harmonic coefficients in 3DGS with a coupled feature attribute $\boldsymbol{f}_i \in \mathbb{R}^{16}$ and utilizes a high-dimensional rendering pipeline to render it as a coupled feature field $\mathbf{F}_{coup} \in \mathbb{R}^{H \times W \times 16}$. Furthermore, we use a public scene decoder and a private message decoder to decouple $\mathbf{I}_{ori} \in \mathbb{R}^{H \times W \times 3}$ and $\mathbf{I}_{hid} \in \mathbb{R}^{H \times W \times 3}$ from $\mathbf{F}_{coup}$, respectively. Finally, we use the combination of $\ell_{ori}$ and $\ell_{hid}$ to constrain the optimization of the learnable attributes of Gaussian points and the weights of two decoders.

Although GS-Hider achieves good rendering quality and the public point cloud file does not exhibit suspicious attributes in terms of format, when we visualize GS-Hider's point cloud, the geometric structure of the hidden information is exposed in the public point cloud. For example, as shown in Fig. 1(b), we observe that when attempting to hide the "mic" in the "bonsai", the shape of the microphone appears in the public point cloud file. This is because GS-Hider uses the same set of points to represent both the original scene and the hidden object, inevitably resulting in geometric and spatial position overlap. Moreover, GS-Hider does not implement an appropriate optimization strategy to control the growth of Gaussian points used to render the hidden message, which exacerbates the compromise of point cloud security.

### 3.2 TASK SETTINGS AND OUR OBJECTIVES

Following the setup of GS-Hider (Zhang et al., 2024b), we aim for the SecureGS framework to maintain transparency and generalization. **Transparency** denotes that after users embed information into the original 3D scene, they can openly publish the container 3DGS for online rendering, while preventing unauthorized users from decrypting the hidden information. **Generalization** refers to the framework's ability to adapt to hiding information in 3D objects, images, and bits. To be noted, unlike GS-Hider, we aim to ensure both the **file format security** and the **geometric structure security** of the container 3DGS point cloud file. Considering that hiding a large-scale 3D scene makes it difficult to ensure that the point cloud structure remains confidential, we only hide a 3D object within the 3D scene. Fig. 2 presents our task settings and realized functions.

### 3.3 HYBRID DECOUPLED GAUSSIAN ENCRYPTION REPRESENTATION

Similar to previous approaches (Lu et al., 2023), we use the sparse point cloud produced by COLMAP as the initial input and voxelize the scene, forming the voxel centers $\mathbf{V} \in \mathbb{R}^{N \times 3}$. Each voxel center $v \in \mathbf{V}$ is treated as an anchor point and equipped with a local context feature $\boldsymbol{f}_v \in \mathbb{R}^{32}$, a scaling factor $\boldsymbol{l}_v \in \mathbb{R}^3$, and two groups of learnable offsets $\{\boldsymbol{O}_{v \oplus i}^{ori}\}_{i=1}^k$ and $\{\boldsymbol{O}_{v \oplus j}^{hid}\}_{j=1}^k$, which respectively generates the dense Gaussian points representing the original scene and hidden object. To avoid the format inconsistency between the container point cloud file and the original Scaffold-GS caused by explicitly storing $\{\boldsymbol{O}_{v \oplus j}^{hid}\}_{j=1}^k$, we design an explicit-implicit hybrid Gaussian encryption representation, as shown in Fig. 3. For the Gaussian points representing the original scene, we explicitly store $k$ learnable offsets $\{\boldsymbol{O}_{v \oplus i}^{ori}\}_{i=1}^k$ for each anchor point $\boldsymbol{x}_v$, and use a scaling factor $\boldsymbol{l}_v$ to determine the position of each Gaussian point $\{\boldsymbol{\mu}_{v \oplus i}^{ori}\}_{i=1}^k$.

$$\{\boldsymbol{\mu}_{v \oplus 0}^{ori}, \ldots, \boldsymbol{\mu}_{v \oplus (k-1)}^{ori}\} = \mathbf{x}_v + \{\boldsymbol{O}_{v \oplus 0}^{ori}, \ldots, \boldsymbol{O}_{v \oplus (k-1)}^{ori}\} \cdot \boldsymbol{l}_v, \tag{3}$$

Additionally, we adopt an implicit neural decoder $\mathcal{F}_o^{\dagger}$ to store the offsets of Gaussian points that render the hidden object. Following (Lu et al., 2023), we also extend $\boldsymbol{f}_v$ to be multi-resolution and

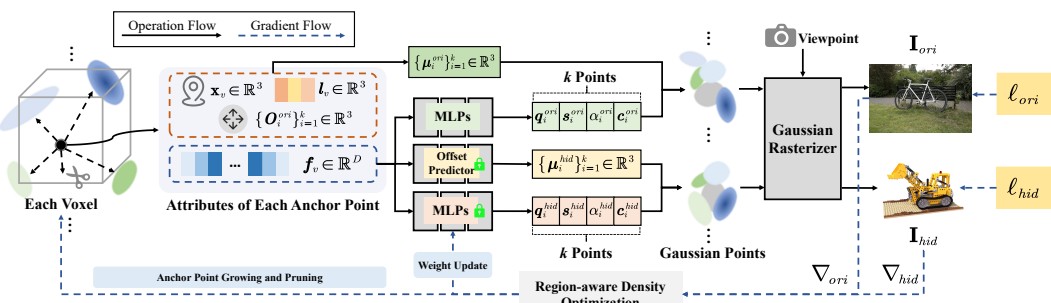

Figure 3: Overall framework of our SecureGS. We first voxelize the scene, where each voxel contains an anchor point with the position $\mathbf{x}_v$, feature $\boldsymbol{f}_v$, scaling factor $\boldsymbol{l}_v$, and offsets $\{\boldsymbol{O}_{v\circledast i}^{ori}\}_{i=1}^k$. Then, we explicitly compute the positions $\boldsymbol{\mu}_{v\circledast i}^{ori}$ via Eq. 3, and predict attributes $\{\boldsymbol{c}_{v\circledast i}^{ori}, \alpha_{v\circledast i}^{ori}, \boldsymbol{q}_{v\circledast i}^{ori}, \boldsymbol{s}_{v\circledast i}^{ori}\}$ via a series of public MLPs. Meanwhile, we use private offset predictor $\mathcal{F}_o^\dagger$ and MLPs to store the position and attributes of the Gaussian points representing the hidden object. Finally, we design a region-aware density optimization to control the Gaussian point growing and pruning.

view-dependent blended feature $\hat{\boldsymbol{f}}_v$. Given the camera located at $\mathbf{x}_c$ and an anchor at $\mathbf{x}_v$, the hidden offsets are computed as follows:

$$\{\boldsymbol{\mu}_{v\circledast 0}^{hid}, \dots, \boldsymbol{\mu}_{v\circledast (k-1)}^{hid}\} = \mathbf{x}_v + \mathcal{F}_o^\dagger(\hat{\boldsymbol{f}}_v, \delta_{vc}, \vec{d}_{vc}), \tag{4}$$

where $\delta_{vc} = \|\mathbf{x}_v - \mathbf{x}_c\|_2$ denotes the relative distance and $\vec{d}_{vc} = (\mathbf{x}_v - \mathbf{x}_c) / \|\mathbf{x}_v - \mathbf{x}_c\|_2$ denotes the viewing direction. The offset predictor $\mathcal{F}_o^\dagger$ will only be accessible to authorized users.

Furthermore, we use a set of public MLPs $\{\mathcal{F}_c, \mathcal{F}_\alpha, \mathcal{F}_q, \mathcal{F}_s\}$ to predict the opacity $\alpha_{v\circledast i}^{ori} \in \mathbb{R}^1$, quaternion $\boldsymbol{q}_{v\circledast i}^{ori} \in \mathbb{R}^4$, scaling $\boldsymbol{s}_{v\circledast i}^{ori} \in \mathbb{R}^3$ and color $\boldsymbol{c}_{v\circledast i}^{ori} \in \mathbb{R}^3$ of the Gaussian points representing the original scene and adopt the private MLPs $\{\mathcal{F}_c^\dagger, \mathcal{F}_\alpha^\dagger, \mathcal{F}_q^\dagger, \mathcal{F}_s^\dagger\}$ to produce $\{\alpha_{v\circledast j}^{hid}, \boldsymbol{q}_{v\circledast j}^{ori}, \boldsymbol{s}_{v\circledast j}^{hid}, \boldsymbol{c}_{v\circledast j}^{hid}\}$ that render the hidden object. We take the color component as an example.

$$\{\boldsymbol{c}_{v\circledast 0}^{ori}, \dots, \boldsymbol{c}_{v\circledast (k-1)}^{ori}\} = \mathcal{F}_c(\hat{\boldsymbol{f}}_v, \delta_{vc}, \vec{d}_{vc}), \{\boldsymbol{c}_{v\circledast 0}^{hid}, \dots, \boldsymbol{c}_{v\circledast (k-1)}^{hid}\} = \mathcal{F}_c^\dagger(\hat{\boldsymbol{f}}_v, \delta_{vc}, \vec{d}_{vc}). \tag{5}$$

Similarly, other attributes of SecureGS can also be produced by these MLPs based on the blended features, the distance, and the viewing direction. Finally, all 3D Gaussian points are rendered into 2D views of the original scene $\mathbf{I}_{ori}$ and the hidden object $\mathbf{I}_{hid}$ via the rasterizer and alpha blending mechanism similar to 3DGS (Kerbl et al., 2023).

## 3.4 REGION-AWARE DENSITY OPTIMIZATION

**Motivation:** Although our hybrid Gaussian encryption mechanism can securely represent the hidden Gaussian points in format, we find that by visualizing the anchor point cloud, the clues of the hidden object can still be discovered from the geometric structure, as shown in Fig. 4. For instance, since the anchor point cloud is relatively sparse, when we hide the 3D object "hot dog" in the "bicycle" scene, the geometric structure of the hot dog is completely exposed, which seriously poses a significant security risk. To

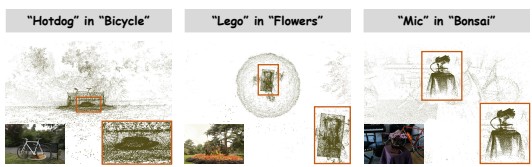

Figure 4: Visualization of the point cloud produced by our SecureGS without region-aware density optimization (RDO) strategy. The RGB reference of the hidden scene is placed on the left bottom.

address this issue, a straightforward solution is to increase the density of Gaussian points, allowing the Gaussian points of the original scene to obscure those of the hidden object. However, this indiscriminate approach of promoting Gaussian point splitting significantly increases the number of anchor points, leading to slower rendering speeds and higher storage costs.

To overcome the trade-off between the storage size and geometry structure security of the container point cloud, we propose a region-aware density optimization strategy to control the splitting and pruning of the Gaussian points adaptively. It can change the threshold of anchor point splitting so that

Gaussian anchor points only grow in large quantities at the location of hidden objects while having a small impact on the rendering and storage efficiency of the overall framework. Specifically, our anchor growing strategy can be divided into three steps: **1) Asynchronous Gradient calculation:** As shown in Fig. 3, for the optimization of both the original scene and the hidden 3D object, we separately accumulate the gradients of the neural Gaussian points involved in backward with different frequencies, denoted as $\nabla_{ori}$ and $\nabla_{hid}$. We aim to slow down the accumulation of $\nabla_{hid}$, thereby suppressing the splitting of the hidden anchor points and enhancing security. **2) Obtain a high-density region:** Since we calculate $\nabla_{ori}$ and $\nabla_{hid}$ separately, we can also perform anchor growing based on $\nabla_{ori}$ and $\nabla_{hid}$ in a decoupled manner. For example, if $\nabla_{hid} > \tau_{fix}$, we will split new anchor points, forming the point set $\Gamma_{hid}$. Note that $\tau_{fix}$ denotes a pre-set splitting threshold. Similarly, we can construct $\Gamma_{ori}$ using $\nabla_{ori} > \tau_{fix}$ as the trigger condition. Afterward, we apply a point cloud clustering algorithm DBSCAN (Ester et al., 1996) on these hidden anchor points $\Gamma_{hid}$ to obtain a bounding box $\mathbb{S}_{bbx}$ where the hidden anchor points are densely distributed. **3) Growing and pruning for $\Gamma_{ori}$ and $\Gamma_{hid}$:** For the growing of $\Gamma_{ori}$, we replace the fixed threshold $\tau_{fix}$ with an adaptive threshold $\tau_{ada}$ for splitting based on whether the points are within the bounding box $\mathbb{S}_{bbx}$.

$$\tau_{ada} = \tau_{fix} \ / \ r_{down}, \text{ if } (x, y, z) \in \mathbb{S}_{bbx} \text{ or } \tau_{fix} \text{ else,} \tag{6}$$

where $(x, y, z)$ denotes the spatial coordinates of an anchor point. $r_{down}$ denotes a gradient down-sampling ratio. Then, we prune some of the anchor points in $\Gamma_{ori} \cup \Gamma_{hid}$ if an anchor fails to pro duce neural Gaussians with a satisfactory level of opacity. Note that we still use $\tau_{fix}$ as the threshold for the growing of $\Gamma_{hid}$ in this stage. The complete algorithm is presented in Alg. 1.

### 3.5 TRANING DETAILS AND LOSS FUNCTIONS

To train the proposed SecureGS, we use the pixel-level $\ell_1$ loss, SSIM term $\ell_{ssim}$ and volume regularization $\ell_{vol}$ (Lu et al., 2023) to optimize the learnable attributes of anchor points and the weights of MLPs $\{\mathcal{F}_c, \mathcal{F}_\alpha, \mathcal{F}_q, \mathcal{F}_s, \mathcal{F}_o^\dagger, \mathcal{F}_c^\dagger, \mathcal{F}_\alpha^\dagger, \mathcal{F}_q^\dagger, \mathcal{F}_s^\dagger\}$. Given the ground truth of the original training view $\hat{\mathbf{I}}_{ori}$ and its corresponding hidden view $\hat{\mathbf{I}}_{hid}$, the supervision is expressed as follows.

$$\ell_{ori} = (1 - \alpha) \cdot \ell_1(\mathbf{I}_{ori}, \hat{\mathbf{I}}_{ori}) + \alpha \cdot \ell_{ssim}(\mathbf{I}_{ori}, \hat{\mathbf{I}}_{ori}) + \beta \cdot \ell_{vol}(\boldsymbol{s}^{ori}), \tag{7}$$

$$\ell_{hid} = (1 - \alpha) \cdot \ell_1(\mathbf{I}_{hid}, \hat{\mathbf{I}}_{hid}) + \alpha \cdot \ell_{ssim}(\mathbf{I}_{hid}, \hat{\mathbf{I}}_{hid}) + \beta \cdot \ell_{vol}(\boldsymbol{s}^{hid}), \tag{8}$$

where $\alpha$ and $\beta$ are used to balance the components of each loss. The volume regularization $\ell_{vol}$ prompts the neural Gaussians to be compact with minimal overlapping via constraining the product of the scale in each Gaussian. Finally, our total loss is $\ell_{total} = \ell_{ori} + \lambda \cdot \ell_{hid}$, where $\lambda$ denotes the trade-off factor to balance the rendering of the original scene and hidden object.

## 4 EXPERIMENTS

### 4.1 EXPERIMENTAL SETUP

**Dataset Preparation:** We respectively conducted experiments on hiding 3D objects, 2D images, and bits in 3D scenes or objects. For 3D object and 2D image hiding, the original scene includes the bicycle (BI.), flowers (FL.), garden (GA.), stump (ST.), treehill (TR.), room (RO.), counter (CO.), kitchen (KI.), bonsai (BO.) from Mip-NeRF 360 (Barron et al., 2021). The hidden 3D object is obtained from the Blender dataset (Mildenhall et al., 2020). We use Supersplat¹ to embed hidden 3D objects into 3D scenes. For bit hiding, we hide 48 bits into the 3D objects (Mildenhall et al., 2020), which is the maximum number of bits that other comparison methods can support.

**Comparison Methods:** For 3D object and 2D image hiding, we compare our SecureGS with existing 3DGS steganography method GS-Hider (Zhang et al., 2024b). Meanwhile, similar to StegaNeRF (Li et al., 2023a), we feed the output of the original 3DGS to a U-shaped decoder and constrain it to output hidden objects, thus implementing a variant called 3DGS+StegaNeRF. For bit hiding, since there is still no 3DGS steganography work for bit hiding available, we compare our method with two SOTA NeRF watermarking methods (Luo et al., 2023b; Song et al., 2024).

**Evaluation Metrics:** We utilize PSNR, SSIM, LPIPS of the original scene, and hidden message to measure the rendering quality of different methods. Meanwhile, FPS and storage size (MB) are

---

¹ https://playcanvas.com/supersplat

Table 1: Comparison of the PSNR(dB), storage size, and FPS performance of the original scenes and hidden message. "Scene-Level" denotes hiding a complete RGB image where the objects are embedded in the original scene, and "Object-level" means hiding isolated objects without a background. The best results are highlighted in pink and the second-best ones are in yellow .

| Method | Type | Size (MB) | FPS | BI. | FL. | GA. | ST. | TR. | RO. | CO. | KI. | BO. | Avg |
|---|---|---|---|---|---|---|---|---|---|---|---|---|---|
| Scaffold-GS | Ori. | 161.46 | 142.91 | 25.01 | 21.26 | 27.19 | 26.49 | 22.96 | 32.26 | 29.48 | 31.35 | 32.61 | 27.62 |
| 3DGS +StegaNeRF | Ori. | 1106.67 | 35.09 | 24.05 | 21.92 | 27.28 | 26.00 | 22.56 | 28.95 | 27.46 | 29.39 | 31.13 | 26.53 |
| | Hid. | | | 29.03 | 29.28 | 32.38 | 27.60 | 28.32 | 31.35 | 27.06 | 32.21 | 31.20 | 29.82 |
| GS-Hider | Ori. | 468.63 | 48.28 | 24.42 | 20.85 | 27.28 | 25.98 | 22.01 | 30.20 | 27.88 | 29.90 | 30.84 | 26.59 |
| | Hid. | | | 30.82 | 28.35 | 32.96 | 28.13 | 27.97 | 34.20 | 27.87 | 32.95 | 32.33 | 30.62 |
| SecureGS (Scene-Level) | Ori. | 267.39 | 131.71 | 25.33 | 21.34 | 27.28 | 26.73 | 22.93 | 32.36 | 29.94 | 31.51 | 32.33 | 27.75 |
| | Hid. | | | 33.74 | 30.45 | 34.47 | 30.24 | 29.38 | 35.93 | 30.11 | 33.21 | 32.99 | 32.28 |
| SecureGS (Object-Level) | Ori. | 290.54 | 106.98 | 25.31 | 21.37 | 27.51 | 26.75 | 22.91 | 32.51 | 30.28 | 31.70 | 33.05 | 27.93 |
| | Hid. | | | 48.04 | 40.70 | 38.23 | 29.77 | 50.96 | 38.45 | 30.77 | 33.22 | 33.75 | 38.21 |

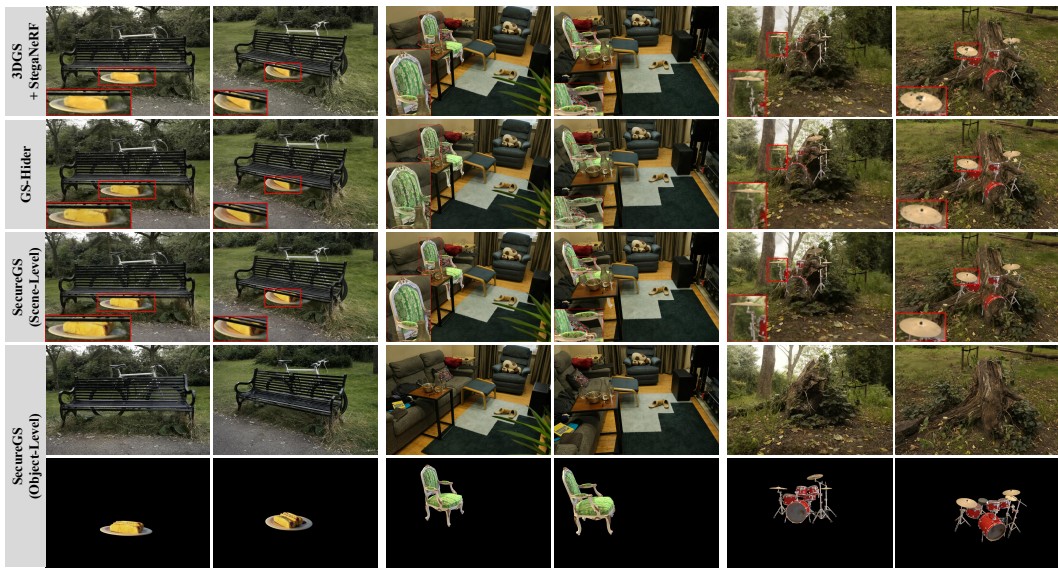

Figure 5: Rendering quality comparison of the hidden message between the proposed SecureGS, previous GS-Hider, and 3DGS+StegaNeRF. For our SecureGS, we also present the decoupled original scene and hidden object on the $4^{th}$ and $5^{th}$ row, which cannot be achieved by other methods.

used to evaluate rendering speed and memory efficiency. Bit accuracy (%) is adopted to verify the decoding precision of our framework.

**Implementation Details:** $\lambda$ is set to 10 when hiding 3D objects and set to 0.1 when hiding a single image. $\alpha$ and $\beta$ in Eq. 8 are respectively set to 0.2 and 0.01. $\tau_{fix}$ and $r_{down}$ are respectively set to 0.0002 and 4. We consistently set $k = 10$ across all experiments and the MLPs used in our approach consist of 2 layers with ReLU activations with each hidden layer having 32 units. We conduct all our experiments on the NVIDIA RTX 4090Ti server and use the same rasterizer as the original 3DGS.

## 4.2 COMPARISON WITH EXISTING 3DGS STEGANOGRAPHY METHODS

To verify the superiority of our method, we compare the proposed SecureGS with two state-of-the-art 3DGS steganography methods in hiding 3D objects in 3D scenes. Our SecureGS has significant advantages in at least the following three aspects.

**Higher rendering fidelity:** As reported in Tab. 5, our method improves the rendering quality of the original scene by 1.16dB and the fidelity of the hidden object by 1.66dB compared to GS-Hider. Furthermore, due to the utilization of more Gaussian points, the PSNR of our rendered original scene can even slightly surpass that of our baseline Scaffold-GS, which proves that our decoupled Gaussian encryption representation ensures the rendering of the hidden object and the original scene do not

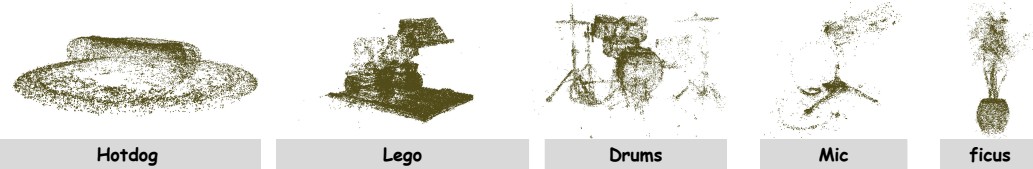

Figure 6: Visualization of the anchor point cloud of our SecureGS produced based on $\nabla_{hid}$.

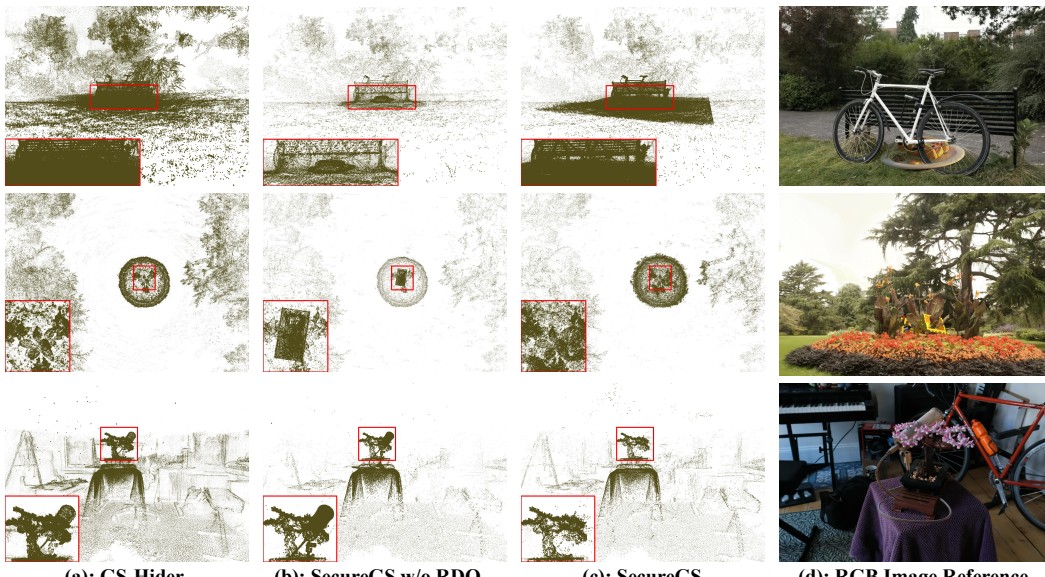

**(a): GS-Hider**  **(b): SecureGS w/o RDO**  **(c): SecureGS**  **(d): RGB Image Reference**

Figure 7: Visualization of the point cloud generated by GS-Hider, SecureGS without RDO strategy, and the proposed SecureGS. Clues of hidden objects cannot be found in our anchor point cloud, while the security of other methods is poor.

interfere with each other. To be noted, since GS-Hider and 3DGS+StegaNeRF decode the hidden message from a feature map or RGB image, they often suffer loss collapse when directly decoding objects with black/white backgrounds. To ensure a fair comparison with previous methods, we retained the content of the original scene when hiding objects, thus forming a complete RGB image as the $\hat{\mathbf{I}}_{hid}$, and we refer to this type of hiding as "scene-level". Meanwhile, "object-level" means that we use isolated 3D objects, without any background, as hidden content $\hat{\mathbf{I}}_{hid}$ for the training of our SecureGS. As shown in Fig. 5, our SecureGS demonstrates a clear advantage in scene-level hiding compared to previous methods, rendering the texture structure of hidden objects more clearly and realistically, better integrating them with the surrounding background, and producing fewer artifacts and noise. These results prove that our method achieves higher rendering quality in both subjective effects and qualitative metrics.

**Lower storage size and faster rendering speed:** As shown in Tab. 1, our SecureGS reduces storage space by $201.24$MB compared to GS-Hider, and its rendering speed is nearly 3 times faster, verifying that our rendering mechanism is significantly more efficient. Compared to the original Scaffold-GS, the FPS of our SecureGS decreases by only 7% at the scene level and 25% at the object level, which is acceptable and still maintains good real-time capability.

**Decoupled original scene and hidden message decoding:** In addition to the above two performance advantages, our SecureGS can decouple the original and hidden RGB views and separate the anchor point clouds produced based on $\nabla_{ori}$ and $\nabla_{hid}$. As shown in Fig. 6, the hidden anchor point decoded by our method retains a good geometry structure and is very close to the input hidden object, which greatly increases the flexibility and practicality of our framework.

### 4.3 SECURITY ANALYSIS

We analyze the security of SecureGS from three aspects. From the view of the point cloud file format, our SecureGS only stores the feature of anchor points $\boldsymbol{f}_v$, along with the offset $\{\boldsymbol{O}_{v\circledast i}^{ori}\}_{i=1}^k$ and

Table 2: Robustness analysis under different pruning ratios. $\text{PSNR}_o$ and $\text{PSNR}_h$ denote the fidelity of original and hidden message.

| Pruning Ratio | $\text{PSNR}_o$ | $\text{SSIM}_o$ | $\text{PSNR}_h$ | $\text{SSIM}_h$ |
|---|---|---|---|---|
| 5% | 27.41 | 0.814 | 37.45 | 0.985 |
| 15% | 26.30 | 0.794 | 35.43 | 0.983 |
| 20% | 24.96 | 0.765 | 33.90 | 0.981 |

Table 3: Rendering quality and bit accuracy comparison between our SecureGS and other competitive methods on Blender dataset.

| Methods | Bit Acc. ↑ | PSNR ↑ | SSIM ↑ | LPIPS ↓ |
|---|---|---|---|---|
| CopyRNeRF | 62.15 | 25.50 | 0.907 | 0.089 |
| NeRFProtector | 92.69 | 29.26 | 0.939 | 0.048 |
| SecureGS | 100.00 | 33.84 | 0.968 | 0.003 |

scaling factor $\boldsymbol{l}_v$ of the Gaussian points used for rendering the original scene, which is identical to the file format of the original Scaffold-GS. From the view of the visualized point cloud structure, as shown in Fig. 7, almost no traces of the hidden scene can be detected in our anchor points, whereas other methods fail to achieve this. For instance, when we hide a microphone in the "bonsai" scene (the third line in Fig. 7), GS-Hider, lacking explicit control of point cloud growing, reveals the shape of the microphone in the visualized point cloud. Similarly, if SecureGS does not employ our region-aware density optimization (RDO), the hidden message would also be exposed due to the use of sparse anchor point clouds. From the view of rendered images, as shown in Fig. 5, the rendered scene does not reveal any artifacts or edges of the hidden object and maintains a high fidelity. Therefore, we can conclude that our SecureGS is secure and reliable.

## 4.4 Robustness Analysis

To evaluate the robustness of our SecureGS, We perform random pruning on the anchor Gaussian points. To be noted, random pruning denotes randomly pruning a proportion of anchor points. PSNR and SSIM results of the original scene ($\text{PSNR}_o$, $\text{SSIM}_o$) and hidden object ($\text{PSNR}_h$, $\text{SSIM}_h$) are reported in Tab. 2. We can find that randomly pruning 5% of anchors has almost no effect on the rendering quality of SecureGS. Even at a larger pruning rate of 25%, our method can still achieve 24.96 dB / 33.90 dB on $\text{PSNR}_o$ / $\text{PSNR}_h$, which verifies that our method is robust enough to the degradation of point clouds. **More visualized results are presented in the appendix.**

## 4.5 Ablation Studies

We conduct ablation studies on two key modules of our SecureGS: hybrid decoupled Gaussian encryption representation(HDGER) and region-aware density optimization (RDO). Since both HGDER and RDO are essential components for ensuring the security of SecureGS, we only focus on exploring how adding these two modules affects the rendering fidelity of the hidden mes-

Table 4: Ablation studies on two key modules of our SecureGS, namely HDGER and RDO.

| Method | Size(MB) | $\text{PSNR}_o$ | $\text{SSIM}_o$ | $\text{PSNR}_s$ | $\text{SSIM}_s$ |
|---|---|---|---|---|---|
| Ours w/o HDGER and RDO | 185.79 | 27.85 | 0.817 | 40.51 | 0.992 |
| Ours w/o HDGER | 254.91 | 27.81 | 0.814 | 38.68 | 0.988 |
| Ours w/o RDO | 168.75 | 27.49 | 0.805 | 40.42 | 0.991 |
| Ours | 290.54 | 27.93 | 0.822 | 38.21 | 0.986 |

sage and the original scene. By removing HGDER, we directly use an explicit $\boldsymbol{O}_{v\circledast j}^{hid}$ to store the offsets of the hidden Gaussian points, referred to as "Ours w/o HGDER". As reported in Tab. 4, "Ours w/o HGDER" has almost no significant impact on the rendering quality of SecureGS. Meanwhile, when RDO is removed, we observe a 2.21dB gain on PSNR of the hidden scene due to the lack of restrictions on the splitting of hidden Gaussian points. However, the rendering quality of the original SecureGS remains satisfactory when weighed against the security enhancements provided by RDO. Removing both modules, although the fidelity of the hidden object further improves, the security of our method is significantly compromised (as shown in Fig. 9 of the appendix).

## 4.6 Extensions

In addition to hiding 3D objects in 3D scenes, our SecureGS is also applicable to hiding bits, and single images in the original scene.

**Hiding bits in 3D Scene**: To achieve efficient hiding and lossless decoding of bit information, we introduce an additional, private MLP $\mathcal{F}_b^\dagger$ based on the original Scaffold-GS. $\mathcal{F}_b^\dagger$ takes the feature $\boldsymbol{f}_v \in \mathbb{R}^{32}$ of each voxel as input to convert the high-dimensional features into a specific bit length. Finally, we use the average of the bit sequences decoded from all voxels as the final copyright. The bit accuracy and rendering quality of our method and other competitive methods are presented in

Table 5: Rendering quality comparison (PSNR (dB)) of the original scene and hidden image.

| Method | Type | Size (MB) | FPS | BI. | FL. | GA. | ST. | TR. | RO. | CO. | KI. | BO. | Avg |
|---|---|---|---|---|---|---|---|---|---|---|---|---|---|
| 3DGS +StegaNeRF | Ori. | 842.51 | 52.94 | 18.44 | 15.95 | 21.93 | 20.86 | 18.25 | 23.94 | 23.21 | 21.58 | 24.52 | 20.96 |
| | Hid. | | | 37.95 | 35.82 | 37.23 | 36.21 | 38.85 | 40.92 | 38.79 | 39.51 | 40.16 | 38.38 |
| GS-Hider | Ori. | 385.41 | 57.35 | 24.38 | 20.74 | 26.84 | 25.91 | 21.92 | 30.49 | 28.75 | 29.72 | 31.05 | 26.64 |
| | Hid. | | | 40.25 | 45.19 | 40.28 | 41.54 | 40.19 | 40.56 | 42.72 | 42.06 | 46.66 | 42.16 |
| SecureGS | Ori. | 155.09 | 144.46 | 24.86 | 21.02 | 27.07 | 26.21 | 22.86 | 31.96 | 29.13 | 30.52 | 31.70 | 27.26 |
| | Hid. | | | 45.40 | 47.11 | 38.87 | 42.99 | 39.70 | 41.27 | 42.95 | 41.92 | 46.76 | 42.99 |

Figure 8: Visualization of the rendered scene and recovered image produced by our SecureGS and other methods. Hidden view denotes the specific view that is used to hide the image.

Tab. 3. Compared to CopyRNeRF (Luo et al., 2023b) and NeRFProtector (Song et al., 2024), we can achieve 100% bit decoding accuracy because we can decode directly from the point cloud and the bits decoded from each voxel can be cross-validated. However, other methods decode bits from the 2D view with certain errors. Meanwhile, after adding bits, the fidelity of our method can still reach $33.84$ dB on PSNR, which far exceeds CopyRNeRF and NeRFProtector. Note that both these methods use a message decoder for image watermarking, they possess unique advantages in the generalization of message extraction.

**Hiding a single image in 3D Scene**: Hiding a single image is a special case of hiding 3D objects. Here, our task is to embed a copyrighted image in a specific view of a 3D scene. Similar to GS-Hider (Zhang et al., 2024b), during the fitting of the original scene, we only encourage the rendering result at this specific view to be close to the hidden image in each iteration, without constraining other views. The results are reported in Tab. 5. It can be seen that our method outperforms GS-Hider by 0.62dB and 0.83dB on the PSNR of the original scene and restored image with a smaller storage space and a faster rendering speed. As plotted in Fig. 8, our rendered original view can achieve more detailed reconstruction, especially in some areas with complex textures such as grass, while GS-Hider often appears blurry in these areas. Meanwhile, 3DGS+StegaNeRF is more inclined to remember the embedded copyright image because it is decoded from the rendered RGB view, which will cause severe aliasing and interference of the rendered view and copyright image.

## 5 CONCLUSION

In this paper, we propose SecureGS, a novel and efficient 3DGS steganography framework. SecureGS successfully addresses the challenges of security and fidelity in previous 3DGS steganography approaches by incorporating anchor point-based neural decoding, a hybrid Gaussian encryption mechanism, and a region-aware density optimization. Our approach allows for secure embedding and retrieval of hidden 3D content, copyright images, and bits, ensuring high fidelity in rendering both the original and hidden message. Extensive experiments confirm that our SecureGS outperforms existing methods in rendering speed, fidelity, and security, showcasing its potential for real-time applications. Furthermore, our work lays a promising foundation for advancements in copyright protection and encrypted transmission of 3D assets, offering a new avenue for safeguarding 3D content in various digital media landscapes.

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

# A  APPENDIX

## A.1  MORE IMPLEMENTATION DETAILS

**Dataset Construction:** To make the training view set of the hidden object and the original scene correspond to each other, we use the Supersplat to align the position of the original scene and hidden object and render the training set of the hidden object according to the viewpoints in the training set of the original scene. The correspondence between the hidden object and original scenes is listed in Tab. 6.

Table 6: Correspondence between hidden objects and original scenes.

| Original Scene | Bicycle | Flowers | Garden | Stump | Treehill | Room | Counter | Kitchen | Bonsai |
|---|---|---|---|---|---|---|---|---|---|
| Hidden Object | hotdog | lego | ficus | drums | ficus | chair | lego | ship | mic |

**Implementation Details and Network Structure:**

Feature bank: We follow the design of the feature bank introduced in Scaffold-GS (Lu et al., 2023) to extend $\boldsymbol{f}_v$ to be multi-resolution and view-dependent feature $\hat{\boldsymbol{f}}_v$. Specifically, for each anchor $v$, we generate a feature bank $\{\boldsymbol{f}_v, \boldsymbol{f}_{v_{\downarrow_1}}, \boldsymbol{f}_{v_{\downarrow_2}}\}$, which is then fused using view-dependent weights to obtain a combined anchor feature $\hat{\boldsymbol{f}}_v$. Given the camera located at $\mathbf{x}_c$ and an anchor at $\mathbf{x}_v$, we compute their relative distance and viewing direction as follows:

$$\delta_{vc} = \|\mathbf{x}_v - \mathbf{x}_c\|_2, \quad \vec{d}_{vc} = \frac{\mathbf{x}_v - \mathbf{x}_c}{\|\mathbf{x}_v - \mathbf{x}_c\|_2}, \tag{9}$$

The feature bank is then blended through a weighted sum, with the weights predicted by a small MLP $\mathcal{F}_w$. The integrated anchor feature $\hat{\boldsymbol{f}}_v$ is then calculated as:

$$\{w, w_1, w_2\} = \text{Softmax}(\mathcal{F}_w(\delta_{vc}, \vec{d}_{vc})), \tag{10}$$

$$\hat{\boldsymbol{f}}_v = w \cdot \boldsymbol{f}_v + w_1 \cdot \boldsymbol{f}_{v_{\downarrow_1}} + w_2 \cdot \boldsymbol{f}_{v_{\downarrow_2}}. \tag{11}$$

The structure of MLPs: The structure of our MLPs follows a "Linear $\rightarrow$ RELU $\rightarrow$ Linear" style with the hidden dimension of 32. The structure of $\{\mathcal{F}_\alpha^\dagger, \mathcal{F}_s^\dagger, \mathcal{F}_q^\dagger, \mathcal{F}_c^\dagger\}$ is identical to $\{\mathcal{F}_\alpha, \mathcal{F}_s, \mathcal{F}_q, \mathcal{F}_c\}$ in Scaffold-GS. Note that for the offset predictor $\mathcal{F}_o^\dagger$, the last linear layer is to transform intermediate tensors from $\mathbb{R}^{N \times 32}$ to $\mathbb{R}^{N \times (3k)}$, where $N$ and $k$ respectively denote the number of anchor points and hidden Gaussian points generated by each voxel.

**The algorithm of our RDO Strategy:** To more clearly demonstrate our region-aware density optimization (RDO) algorithm, we have provided the pseudocode in Alg. 1.

## A.2  RELATIONSHIPS AND DIFFERENCES WITH EXISTING 3D STEGANOGRAPHY METHODS

**Comparison with GS-Hider:** Our SecureGS and GS-Hider (Zhang et al., 2024b) follow similar task setups, assuming that the point cloud file needs to be publicly available while hiding information. At a high level, SecureGS and GS-Hider adopt a "coupling-decoupling" process. **The main differences** between them lie in the implementation architecture: First, GS-Hider is built on the original 3DGS and embeds information through coupled feature attributes, using a CNN-based neural network to decode the feature maps. The decoding mechanism from feature maps is the root cause of its deficiencies in security, fidelity, and rendering speed. In contrast, our SecureGS is built on the efficient Scaffold-GS, which naturally generates Gaussian points from anchor points specifically for rendering hidden information. Additionally, we use several MLPs to efficiently and in parallel decode the anchor point cloud in batches. Thanks to our efficient mechanism of decoding from anchor points and the proposed region-aware density optimization, we outperform GS-Hider in all aspects. Our method improves fidelity by over 1dB, renders scenes twice as fast as GS-Hider, and requires less storage space while ensuring both the format and geometric structure security of the point cloud files. Additionally, GS-Hider does not support bit embedding and decoding, a limitation that we have addressed in our approach.

**Comparison with HiDDeN-based Methods:** Most previous SOTA 3D watermarking methods (Luo et al., 2023b; Jang et al., 2024a;b; Song et al., 2024) typically use a watermark decoder pre-trained

Table 7: Comparison of 3D scene hiding between GS-Hider and SecureGS.

| Method | $PSNR_o$(dB) | $SSIM_o$ | $LPIPS_o$ | $PSNR_s$(dB) | $SSIM_s$ | $LPIPS_s$ |
|---|---|---|---|---|---|---|
| GS-Hider | 25.82 | 0.783 | 0.246 | 25.18 | 0.780 | 0.306 |
| SecureGS | 27.20 | 0.796 | 0.235 | 29.75 | 0.858 | 0.211 |

on the image domain (*e.g.*, HiDDeN (Zhu et al., 2018)). This watermark decoder is tasked with extracting bits from rendered views by NeRF or 3DGS, offering excellent generalization in message extraction. In contrast, our SecureGS aims to decode bit information directly from the point cloud, which aligns better with the explicit representation features of 3DGS and is more direct and efficient. Currently, there are no deep networks capable of embedding and decoding bits from point clouds, which makes it infeasible for us to adopt HiDDeN-based methods for generalized message extraction. However, our message extraction MLP is lightweight enough that it does not increase training time or cost. We plan to explore the development of a generalized method for extracting bits directly from point clouds in future work.

### A.3 LIMITATIONS AND FUTURE WORKS

**Limitations:** First, to enhance the geometric security of our method, we have to promote anchor point splitting to some extent through our RDO strategy, which results in slightly larger storage space compared to the original Scaffold-GS when hiding 3D objects. Second, to better ensure the security of our steganography framework, the hidden object is required to partially overlap with the point cloud of the original scene to some extent. Hiding a 3D object without leaving traces in areas where the original point cloud is sparse is extremely difficult and remains an area for further exploration.

**Future works:** Our future work will focus primarily on improving the robustness of 3DGS steganography, ensuring that the hidden object's integrity and fidelity are preserved even under more severe damage and aggressive pruning strategies. Additionally, we will explore leveraging 3D backbone networks, such as PointFormer (Pan et al., 2021), to directly decode the hidden point cloud from the original Gaussian points, further enhancing the security of 3D steganography through structured and network-based approaches.

### A.4 RESULTS ON HIDING 3D SCENE INTO 3D SCENE

Following the setup of GS-Hider, we compare our SecureGS with GS-Hider on hiding 3D scenes into 3D scenes. Here, we do not use the bounding box in the RDO strategy, as it is not meaningful for high-capacity scene hiding. The results are reported on Tab. 7. It can be observed that the rendering fidelity of the original scene and the hidden scene in our method is 1.38 dB and 4.57 dB higher than that of GS-Hider, respectively, demonstrating a significant advantage.

### A.5 MORE ROBUSTNESS ANALYSIS

To further validate the robustness of our method, we apply several 3D degradations to the anchor point cloud, including Gaussian noise of varying intensities and point cloud denoising. The metrics are reported on Tab. 8. The Gaussian noise intensities are set to 0.05, 0.1, and 0.15, and the denoising method is Statistical Outlier Removal. It can be observed that our method demonstrates strong resistance to Gaussian noise. Moreover, when dealing with point cloud denoising, although the original scene's rendering quality decreases, the hidden object's rendering quality remains almost unchanged. Fig. 10 further illustrates the robustness of our method under different degradations.

### A.6 MORE ABLATION STUDIES

To further clarify the independent contributions of our region-aware density optimization (RDO), we further conduct ablation studies on RDO and realize two variants, namely globally reducing the gradient threshold and using the same gradient to grow anchor points representing both the original and hidden scenes. The results are reported on Tab. 9. We find that globally lowering the gradient threshold increases the memory size by 114.76 MB, significantly reducing rendering speed

Table 8: PSNR (dB) and SSIM under different degradation conditions.

| Condition | $PSNR_o$ | $SSIM_o$ | $PSNR_s$ | $SSIM_s$ |
|---|---|---|---|---|
| Clean | 27.93 | 0.822 | 38.21 | 0.986 |
| Gaussian noise ($\sigma$=0.05) | 27.87 | 0.820 | 37.52 | 0.984 |
| Gaussian noise ($\sigma$=0.1) | 27.74 | 0.817 | 37.04 | 0.983 |
| Gaussian noise ($\sigma$=0.15) | 27.55 | 0.857 | 36.18 | 0.981 |
| Point cloud denoising | 24.02 | 0.794 | 38.20 | 0.986 |

Table 9: Ablation study on different key steps in the region-aware density optimization.

| Method | Memory Size (MB) | $PSNR_o$ (dB) | $SSIM_o$ | $PSNR_s$ (dB) | $SSIM_s$ |
|---|---|---|---|---|---|
| Globally reducing the gradient threshold | 405.27 | 28.11 | 0.831 | 38.36 | 0.987 |
| Using the same gradient for anchor growing | 252.49 | 27.73 | 0.815 | 37.95 | 0.977 |
| SecureGS (Ours) | 290.54 | 27.93 | 0.822 | 38.21 | 0.986 |

while only resulting in a 0.18 dB gain in PSNR. Additionally, if a shared gradient is used instead of our asynchronous gradient accumulation, the rendering of the original scene and the hidden object interferes with each other, leading to a reduction in rendering quality for both and failing to ensure security. Thus, our region-aware density optimization achieves a well-balanced trade-off between rendering quality, security, and rendering efficiency.

Furthermore, we also present the visualized results of point cloud produced by our baseline "Ours w/o RDO and HDGER" in Fig. 9. "Ours w/o RDO and HDGER" denotes directly adding the offset $O_{v \circledast j}^{hid}$ to the point cloud file and using a series of encrypted MLPs to predict the attributes of the hidden Gaussian points. Meanwhile, the anchor growing and reduction strategy is the same as Scaffold-GS. Quantitative metrics are reported in Tab. 4. Although this method achieves acceptable fidelity for both the hidden object and the original scene, its point cloud completely exposes the information of the hidden object, failing to meet the security requirements of 3DGS steganography. This indicates that simply combining the framework design of GS-Hider with the rendering method of Scaffold-GS cannot achieve steganography with strong security.

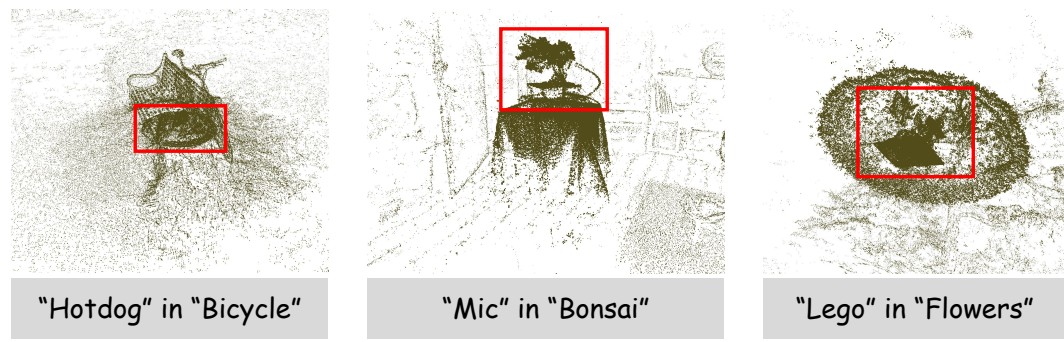

"Hotdog" in "Bicycle"    "Mic" in "Bonsai"    "Lego" in "Flowers"

Figure 9: Visualized results of the point cloud produced by "Ours w/o RDO and HDGER".

## A.7 MORE VISUALIZATION RESULTS

We present more visualization results of our rendering results under different degradations in Fig. 11, the rendering results of bit hiding in Fig. 12, the rendering results of single image hiding in Fig. 13, and the results of 3D object hiding in Fig. 14.

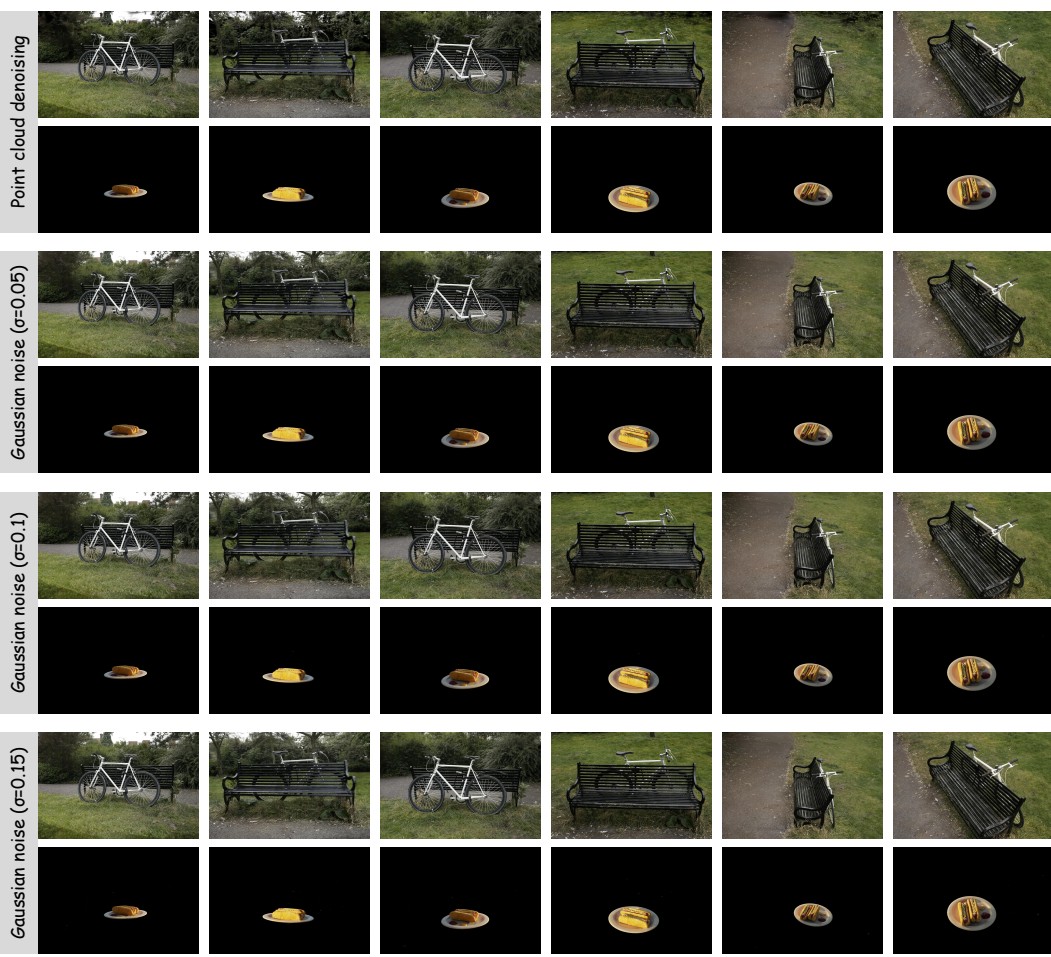

Figure 10: Rendered results of the original scene and hidden object produced by our SecureGS under Gaussian noise and point cloud denoising.

---

**Algorithm 1** Overall Pipeline of Our Proposed Region-Aware Density Optimization

---

$i \leftarrow 0$          ▷ Iteration Count
$\Gamma_{ori} \leftarrow$ SfM points       ▷ Initalization of the anchor point set produced based on $\nabla_{ori}$
$\Gamma_{hid} = \{\}$       ▷ Initalization of the anchor point set produced based on $\nabla_{hid}$
**while** not converged **do**
    $\ell_{total} = \ell_{ori} + \lambda \ell_{hid}$        ▷ Computed via Eq. 8
    $\nabla_{ori} \leftarrow$ AccumulateGradient($\nabla \ell_{total}$),
    **if** $i \% step == 0$ **then**:
        $\nabla_{hid} \leftarrow$ AccumulateGradient($\nabla \ell_{total}$),      ▷ Accumulate gradients in different intervals
    **end if**
    **if** IsRefinementIteration($i$) **then**
        **if** IsBoundingBoxIteration($i$) **then**
            $\mathbb{S}_{bbx} \leftarrow$ PointCloudCluster($\Gamma_{hid}$) ▷ Using the clustering method to get a bounding box
        **end if**
        **for all** anchor points $v(\mathbf{x}_v, \boldsymbol{f}_v, \boldsymbol{l}_v, \boldsymbol{O}_v^{ori})$ in $\Gamma_{ori} \cup \Gamma_{hid}$ **do**
            **if** $v \in \mathbb{S}_{bbx}$ **then**
                $\tau_{ada} = \tau_{fix} / r_{down}$      ▷ Lower the threshold if the anchor in the bounding box
            **end if**
            **if** $\nabla_{ori} > \tau_{ada}$ **then**
                $\Gamma_{ori} \leftarrow$ AnchorGrowing($\Gamma_{ori}$)
                $\Gamma_{ori}, \Gamma_{hid} \leftarrow$ AnchorPruning($\Gamma_{ori}, \Gamma_{hid}$)
            **end if**
            **if** $\nabla_{hid} > \tau_{fix}$ **then**
                $\Gamma_{hid} \leftarrow$ AnchorGrowing($\Gamma_{hid}$)
            **end if**
        **end for**
    **end if**
    $i \leftarrow i + 1$
**end while**

---

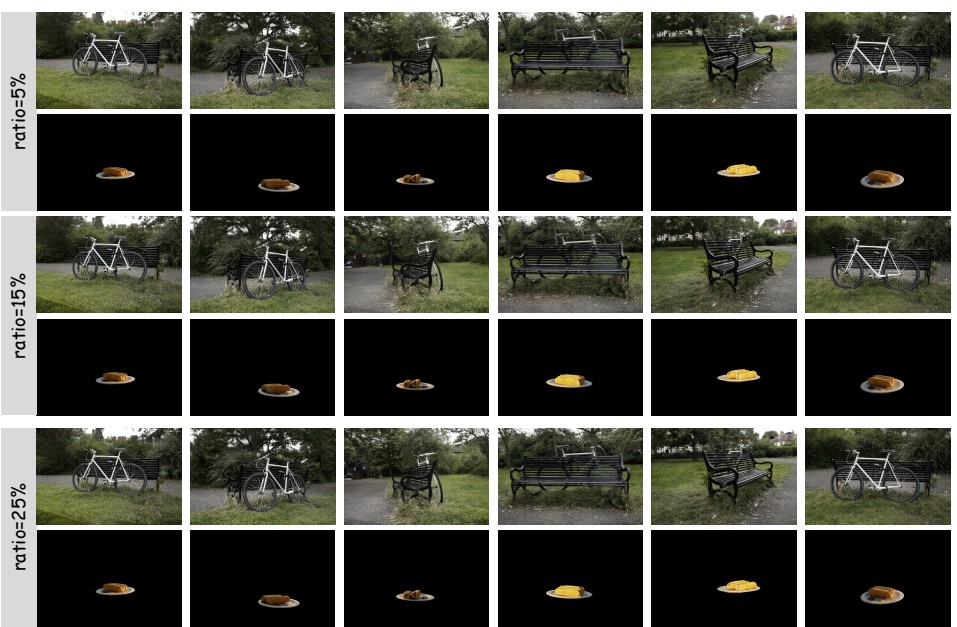

Figure 11: Rendered results of the original scene and hidden object produced by our SecureGS under different degradations.

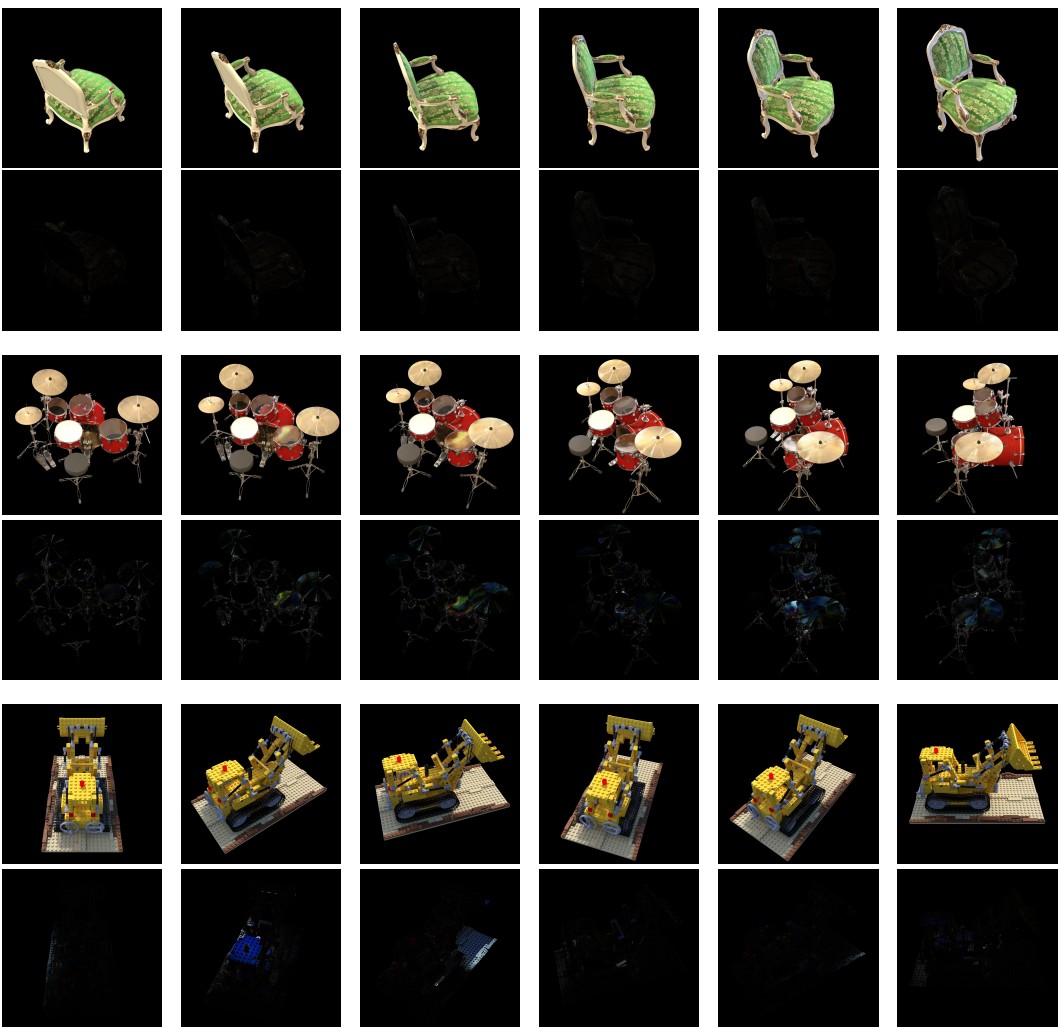

Figure 12: Rendered object and error maps produced by the proposed SecureGS on the blender dataset. After embedding the bit message, the rendering quality of our original scene will hardly be affected.

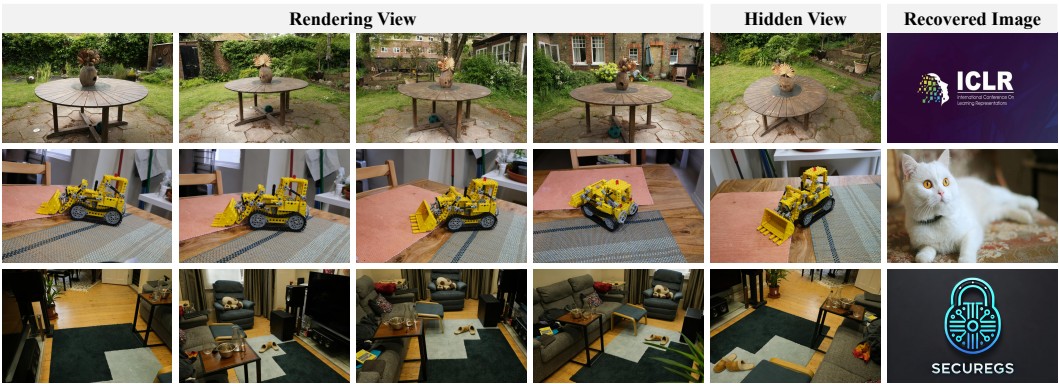

Figure 13: Rendered view and recovered different copyright images produced by our SecureGS.

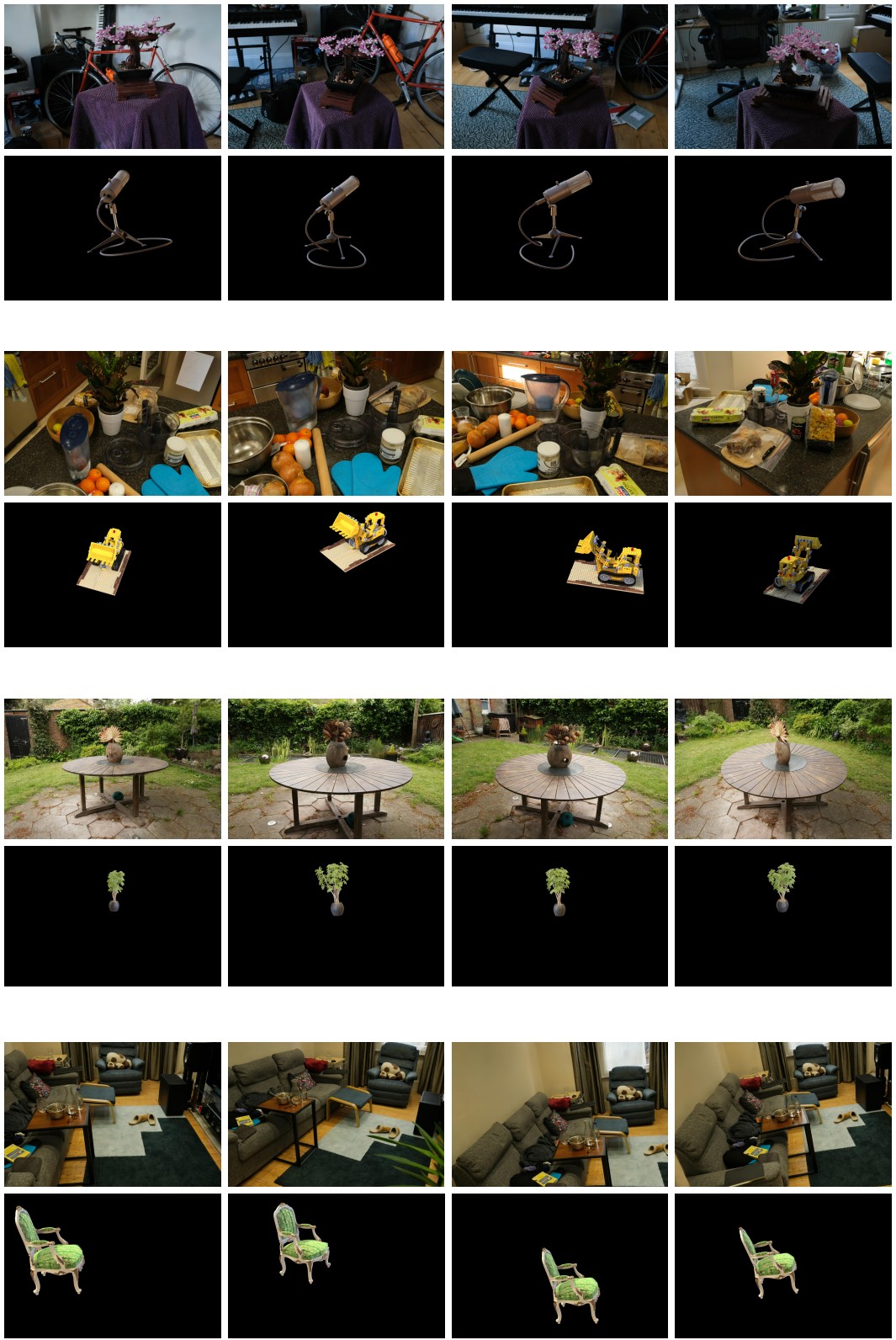

Figure 14: Rendered original scene and the hidden object produced by our SecureGS. The first row of each group is the original scene and the second row is the hidden object.

