# OpenReview forum: "SecureGS: Boosting the Security and Fidelity of 3D Gaussian Splatting Steganography"
_ICLR.cc/2025/Conference — ICLR 2025 Poster_

### Official Review · Reviewer_yt9u · 2024-10-28

**Soundness:** 2
**Presentation:** 3
**Contribution:** 1
**Rating:** 5
**Confidence:** 5

**Summary:**

This paper proposes SecureGS, a 3D Gaussian splatting steganography framework designed to improve fidelity and security in 3D scene steganography. SecureGS combines features from GS-Hider and Scaffold-GS, introducing a hybrid decoupled Gaussian encryption representation where hidden information and original scenes are separately encoded through anchor points. Additionally, a region-aware anchor growth and pruning strategy is introduced to optimize anchor density, enhancing the concealment of hidden information and rendering quality. The experimental results show SecureGS outperforms GS-Hider in rendering speed, fidelity, and security.

**Strengths:**

1. SecureGS effectively combines the strengths of Scaffold-GS and GS-Hider, leveraging anchors and region-aware density optimization to achieve better fidelity and security in information hiding.
2. The region-aware anchor growth and pruning strategy successfully prevents hidden information from being exposed in the public point cloud by adaptively optimizing anchor density.
3. SecureGS shows improved rendering quality, fidelity, and speed compared to GS-Hider and 3DGS+StegaNeRF, particularly in reducing interference between hidden content and original scenes.

**Weaknesses:**

1. SecureGS essentially combines elements of GS-Hider and Scaffold-GS, with limited originality. The core method of SecureGS is to create separate anchor points for the hidden objects and the original scene, which is quite similar to GS-Hider’s approach of creating Gaussian points for both separately—only differing in terms of abstraction level, from points to anchors. While SecureGS introduces a region-aware anchor growth and pruning strategy, these improvements are mostly about implementation details rather than any fundamental innovation. Moreover, the anchor growth and pruning strategies resemble those seen in previous 3D scene optimization work, which makes it hard to classify them as novel contributions. Consequently, the theoretical advancement of SecureGS compared to GS-Hider and Scaffold-GS is not adequately demonstrated, and the overall method feels more like an incremental combination of existing techniques rather than a breakthrough, resulting in relatively weak innovation and technical contributions.

2. The motivation for using Scaffold-GS as the foundational framework is insufficiently explained. Although the paper mentions that using anchor points to generate Gaussian points helps keep their attributes and positions secure since they do not need to be directly stored, these advantages are primarily about rendering efficiency and scalability, rather than any inherent suitability for steganography or encryption tasks. Therefore, directly equating the properties of Scaffold-GS with the encryption encoder-decoder in steganography feels like a logical leap without rigorous backing. The authors need to clarify how Scaffold-GS specifically offers unique advantages for the steganography task and strengthen the rationale for this choice.

3. The ablation studies are too simplistic and fail to convincingly demonstrate SecureGS's unique contributions. Despite claims of outperforming other methods in terms of visual quality (such as PSNR) and file size, the reasons for these improvements are insufficiently explained. As shown in Table 1, much of the performance gain could simply be attributed to the foundation provided by Scaffold-GS, rather than any innovations introduced in SecureGS. Since Scaffold-GS is an upgraded version of Gaussian Splatting, which inherently improves rendering quality and efficiency, directly comparing SecureGS (based on Scaffold-GS) to GS-Hider (based on 3DGS) is insufficient to prove the superiority of the proposed method. The authors need more comprehensive ablation studies to clarify the independent contributions of each new module to ensure that the observed improvements come from their own innovations rather than from Scaffold-GS's inherent advantages. Otherwise, it is difficult to consider these enhancements as unique contributions of SecureGS.

4. SecureGS heavily relies on the Scaffold-GS framework, particularly in the way anchor generation and density optimization are integrated. This deep coupling makes SecureGS less flexible and not readily applicable to other Gaussian splatting models.

5. The effectiveness of the region-aware anchor growth and pruning strategy lacks systematic theoretical support. There is no formal proof or analysis provided to demonstrate whether this strategy is optimal for steganography tasks. Section 3.4's gradient handling and anchor splitting thresholds are mostly based on empirical choices rather than rigorous theoretical derivation. Without a theoretical foundation, the robustness and generalizability of these strategies across different scenes are uncertain, which reduces the reliability and stability of the proposed method in real-world applications.

6. The robustness experiments in SecureGS are insufficient, as they only consider the impact of random anchor pruning, neglecting other significant attack scenarios that could affect the security of steganography. Robustness is critical in evaluating steganographic methods, yet SecureGS lacks tests on important attacks such as point cloud denoising, random point deletion or addition, and geometric transformations. These factors are as crucial as visual quality in the domain of steganography and watermarking, and without proper evaluation of these aspects, the practical security of SecureGS remains questionable.

7. In Table 1, SecureGS shows a significant increase in PSNR compared to Scaffold-GS, which the authors attribute to increased anchor density and the decoupling strategy. However, it seems questionable that the PSNR for the original scene is higher despite SecureGS embedding additional hidden information. Scaffold-GS has no additional hidden data and can focus solely on rendering the scene, while SecureGS introduces extra information that could potentially interfere with the fidelity of the original scene. There is no sufficient core optimization provided in the paper to justify this improvement. This effect seems more likely due to increased anchor density and computational resources rather than a genuine methodological advancement in balancing steganography with scene fidelity. More theoretical analysis or experimental validation is recommended to clarify this observation, ensuring that the PSNR improvements are due to actual innovation rather than just additional resource allocation.

**Questions:**

See weaknesses.

By the way, I do have a small question, purely out of curiosity and not related to the rating: If we perform point cloud downsampling on the anchor points encrypted by SecureGS, would the hidden scene become more apparent in the resulting sparse point cloud? Could it result in a situation similar to GS-Hider as shown in Fig. 7?

---

> ### Author Response · Authors · 2024-11-21
> **Response to Reviewer yt9u (Part 1)**
>
> We sincerely appreciate your detailed and thoughtful review. We have addressed all your concerns, and your feedback has helped make our experiments more comprehensive and our presentation clearer. If you have any further questions, please feel free to continue the discussion with us.
>
> > **W1: Originality of our SecureGS**
>
> - SecureGS and GS-Hider have different method design, theoretical basis, and functionality.
>
>   - Each Gaussian point in GS-Hider is associated with a coupled feature attribute $\boldsymbol{f}_i$, which generates a coupled feature field. Two separate decoders are then used to extract the hidden message and the original scene. In other words, each Gaussian point in GS-Hider is coupled together and contributes to the rendering of both the original scene and the hidden scene, meaning $\textcolor{red}{\textbf{GS-Hider can not achieve creating Gaussian points separately}}$.
>
>   - In contrast, SecureGS achieves a genuine decoupling of Gaussian points for rendering the original scene and the hidden scene by using an implicit offset predictor and a series of attribute MLPs. The anchor point cloud extracted in **Figure 6**, representing the hidden scene, clearly shows this decoupling capability. Thus, the idea of creating separate anchor points for the hidden objects and the original scene $\textcolor{red}{\textbf{is entirely novel and first introduced by SecureGS}}$.
>   - The theoretical basis of GS-Hider is 3DGS, which is a fully explicit representation. In contrast, SecureGS is built on neural voxels and implicit attribute prediction, representing a hybrid of explicit and implicit 3D representation theory.
>
>   - In addition to the functionality of GS-Hider, SecureGS offers several unique and valuable features, such as directly extracting bits from GS points, explicitly separating anchor point clouds used to represent the hidden scene, and hiding background-free objects within the scene. These capabilities clearly distinguish our work from GS-Hider.
>
> - In our region-aware density optimization, we innovatively introduce an asynchronous gradient accumulation strategy, an adaptive bounding box acquisition method, and a decoupled growing and pruning strategy. Together, these enhancements significantly improve the security of 3DGS-based hiding. To the best of our knowledge, such strategies have not been employed in other methods. **If you find a similar approach, please let us know, and we will compare it accordingly.**
>
>
>
> > **W2: Motivation for using Scaffold-GS**
>
> Thank you for your suggestion. Our motivation for choosing Scaffold-GS is as follows:
>
> - **First**, unlike the purely explicit representation of 3DGS, Scaffold-GS uses implicit MLPs to store the attributes of Gaussian points. This approach allows us to focus on encrypting the series of implicit MLPs rather than dealing with a large amount of explicit Gaussian points. This inherent characteristic makes Scaffold-GS particularly well-suited for steganographic tasks.
> - **Second**, Scaffold-GS achieves comparable rendering quality and real-time performance to 3DGS. This means that using Scaffold-GS as a baseline for exploring 3D steganography does not compromise the high fidelity and fast rendering advantages of 3DGS.
>
> We have included this discussion in **Line 73-77 of the main paper**.

---

> ### Author Response · Authors · 2024-11-21
> **Response to Reviewer yt9u (Part 2)**
>
> > **W3: Ablation studies**
>
> We must emphasize that, although our method is built upon Scaffold-GS, **Scaffold-GS itself cannot achieve secure and effective 3DGS steganography**. It is our improvements that enable it to attain enhanced security, high rendering quality and optimized memory usage. We will provide a detailed explanation and include additional ablation experiments to prove it.
>
> - The original Scaffold-GS directly stores the offset of Gaussian points in the point cloud file. If the points used for rendering hidden messages are also stored explicitly, it would expose the geometric structure of the hidden objects. To address this, SecureGS employs an offset predictor combined with a hybrid decoupled Gaussian encryption representation (HDGER), enhancing its security. The fidelity results are presented in **Table 4 of the main paper**, labeled as ''Ours w/o HDGER.''
>
> - To further clarify the independent contributions of our region-aware density optimization (RDO),  we conduct ablation studies on RDO and realize two variants, namely **globally reducing the gradient threshold** and **using the same gradient to grow anchor points representing both the original and hidden scenes**. The results are reported as follows. We find that globally lowering the gradient threshold increases the memory size by 114.76 MB, significantly reducing rendering speed while only resulting in a 0.18 dB gain in PSNR. Additionally, if a shared gradient is used instead of our asynchronous gradient accumulation, the rendering of the original scene and the hidden object interfere with each other, leading to a reduction in rendering quality for both and failing to ensure security. Thus, using our region-aware density optimization achieves a well-balanced trade-off between rendering quality, security, and rendering efficiency. We include this part in **A.6 of our appendix**.
>
>   | Method                                     | Memory Size (MB) | PSNR_o (dB) | SSIM_o | PSNR_s (dB) | SSIM_s |
>   | :----------------------------------------- | :--------------: | :---------: | :----: | :---------: | :----: |
>   | Globally reducing the gradient threshold   |      405.27      |    28.11    | 0.831  |    38.36    | 0.987  |
>   | Using the same gradient for anchor growing |      252.49      |    27.73    | 0.815  |    37.95    | 0.977  |
>   | SecureGS (Ours)                            |      290.54      |    27.93    | 0.822  |    38.21    | 0.986  |
>
>
>
> > **W4: Relies on the Scaffold-GS framework**
>
> The core of our method lies in using implicit MLPs to hide the attributes of Gaussian points, thereby enabling the hiding of 3D objects, images, and bits. Our method only requires adding an embedding attribute $\boldsymbol{f}_v\in\mathbb{R}^{D}$ to any Gaussian representation, making it compatible with our approach. Therefore, SecureGS is a general framework that can integrate with various 3DGS variants, not limited to Scaffold-GS.
>
>
>
> > **W5: Systematic theoretical support and empirical choices of hyper-parameters**
>
> - In the field of 3DGS, it is very rare and challenging to prove the effectiveness of a method via mathematical theoretical derivations. However, we have demonstrated the superiority of SecureGS through detailed experimental results.
> - Our method is parameter-insensitive. **First**, for the same scene, selecting different parameters for RDO, such as the splitting threshold or the iteration at which RDO begins, has minimal impact on the results. **Second**, the same set of parameters can be applied to different scenes without requiring manual adjustments, ensuring the practicality of our approach.
>
>
>
> > **W6: More robustness experiments**
>
> Thanks for your valuable advice. We have supplemented the robustness analysis of point cloud denoising as follows. After applying a Statistical Outlier Removal to denoise the Gaussian point cloud, we find that while the quality of the original scene decreased, the rendering quality of the hidden object remains almost unchanged. This shows the strong robustness of our method. **Subjective results are provided in Figure 9 of our appendix**. Additionally, we have included experiments where Gaussian noise was added to the anchor point cloud in **A.5 of our appendix**.
>
> | Method                | PSNR_o (dB) | SSIM_o | PSNR_s (dB) | SSIM_s |
> | :-------------------- | :---------: | :----: | :---------: | :----: |
> | Point cloud denoising |    24.02    | 0.794  |    38.20    | 0.986  |
> | SecureGS (Ours)       |    27.93    | 0.822  |    38.21    | 0.986  |

---

> ### Author Response · Authors · 2024-11-21
> **Response to Reviewer yt9u (Part 3)**
>
> > **W7: The source of PSNR improvements over Scaffold-GS**
>
> We kindly remind you that we have never stated that SecureGS achieves better fidelity than Scaffold-GS. This is because, in steganography tasks, as the container hides more information, its quality will inevitably be lower than the original carrier, whether in 3DGS or image steganography. In the main paper, we simply aimed to explain this counterintuitive phenomenon: SecureGS shows a slightly higher fidelity (0.13 dB) than Scaffold-GS for the original scene because Scaffold-GS uses fewer Gaussian points, whereas SecureGS, by lowering the splitting threshold in key regions, generates more Gaussian points, resulting in a slight fidelity increase. **Thus, this is not an advantage of our method, but merely an explanation of this experimental result.**
>
>
>
> > **Q1:  Would the hidden scene become more apparent in the resulting sparse point cloud?**
>
> - Due to the introduction of RDO, which encourages anchor splitting, the points representing the original scene and the hidden object are very compact and close to each other. Thus, after downsampling, the information of the hidden object remains disordered and invisible.
> - If an excessively high downsampling ratio is applied, it will severely affect the rendering quality of the original scene, damaging the entire 3DGS. This makes attacks on our steganography algorithm relatively meaningless.

---

> > ### Comment · Reviewer_yt9u · 2024-11-26
> > **Thank you for your efforts**
> >
> > Thank you for your response. The doubts I raised in Weaknesses 2, 6, and 7 have been well addressed. However, the other parts of your response did not fully satisfy me.
> >
> > 1. Over the past few days, I carefully revisited the Gaussian splatting papers I’ve read in the last six months. I am confident I came across something similar to the RDO approach, but I cannot locate it at the moment. I will set this issue aside for now. However, I still believe this paper is essentially a combination of GS-Hider and Scaffold-GS, with two minor components added to make it work. Most of the contributions still stem from GS-Hider and Scaffold-GS, leaving the paper’s innovative contributions quite limited.
> >
> > 3. The authors stated in their rebuttal:
> >    > "We must emphasize that, although our method is built upon Scaffold-GS, Scaffold-GS itself cannot achieve secure and effective 3DGS steganography."
> >
> >    However, 3DGS itself is not designed for 3D steganography—that is precisely the purpose of GS-Hider. The primary contribution of the authors lies in coupling GS-Hider and Scaffold-GS, followed by the introduction of the RDO and HDGER strategies. Therefore, the baseline the authors should have used is an improved version of GS-Hider+Scaffold-GS, which is essentially `Ours w/o RDO and HDGER`. Unfortunately, I believe the current ablation experiments are still flawed.
> >
> > 4 & 5. I remain skeptical about these points. Based on the authors’ explanation, I anticipate that the performance of this method will drop significantly once it is detached from the anchor framework of Scaffold-GS. Additionally, in my experience, the scene size significantly influences parameter choices in Scaffold-GS, and incorrect parameters can lead to drastically different outcomes. However, without the code or supporting materials, there is no way to determine whether my claims or the authors’ are correct. Therefore, I will not factor these points into my scoring.
> >
> > In conclusion, I appreciate the authors' efforts. This is an interesting paper, but many areas still require improvement. In my opinion, the authors must articulate their contributions more clearly, design their experiments more carefully and correctly, and provide more theoretical analysis. For now, I will not raise my score.
> >
> > That said, if the authors release their code and I am able to verify that Weaknesses 4 and 5 are indeed as the authors claim, I would be willing to increase my score.

---

> > > ### Author Response · Authors · 2024-11-27
> > > **Response to the remaining concerns**
> > >
> > > Thanks for your prompt response and recognition of our work. Regarding your remaining concerns, we would like to explain as follows.
> > >
> > > > **W1:  About our novelty and contribution**
> > >
> > > - The main innovation of our method lies in the discovery of a significant issue in existing steganography methods, where visualizing the point cloud can easily leak the geometric information of the hidden object. To address this, we ensure that the structure of the point cloud file remains consistent with the original Scaffold-GS via "HDGER", **thereby ensuring File Format Security**. Additionally, we use RDO to control the adaptive splitting of anchor points, ensuring **Geometric Structure Security**. This improvement in security greatly enhances the practicality of existing steganography methods.
> > > - **The key reason for the fidelity improvement in SecureGS lies in the introduction of implicit neural Gaussian decoding**. This approach allows SecureGS to completely decouple the representation of the original scene and the hidden object. In contrast, each Gaussian point in GS-Hider is responsible for representing both, leading to severe aliasing. Thus, **the introduction of implicit neural Gaussian decoding in GS steganography is another important contribution of our work**.
> > >
> > >
> > >
> > > > **W2: Ablation Study**
> > >
> > > - Following your valuable advice, we realize a baseline "Ours w/o RDO and HDGER". It denotes directly adding the offset $\boldsymbol{O}_{v\circledast j}^{hid}$ to the point cloud file and using a series of encrypted MLPs to predict the attributes of the hidden Gaussian points. Meanwhile, the anchor growing and reduction strategy is the same as Scaffold-GS. Quantitative metrics are reported below (**See Tab.4 in the main paper**).
> > >
> > >   | Model                  | PSNR_o (dB) | SSIM_o | PSNR_s (dB) | SSIM_s |
> > >   | :--------------------- | :---------: | :----: | :---------: | :----: |
> > >   | Ours w/o RDO and HDGER |    27.85    | 0.817  |    40.51    | 0.992  |
> > >
> > >
> > >
> > > - Although this method achieves acceptable fidelity for both the hidden object and the original scene, its point cloud completely exposes the information of the hidden object, failing to meet the security requirements of 3DGS steganography (**See Fig.9 in the Appendix**). This indicates that simply combining the framework design of GS-Hider with the rendering method of Scaffold-GS cannot achieve steganography with strong security.
> > >
> > >
> > >
> > > > **W4 & W5：Applicable to other 3D rendering methods and hyperparameter setup**
> > >
> > > We have released our code in the **supplementary materials**. We will further explain these two points.
> > >
> > > - Our method is adaptable to other frameworks because “HDGER” and “RDO” are two plug-and-play modules. For the original 3DGS, we only need to replace the original spherical harmonic coefficients with a feature embedding, and then use several MLPs to decode attributes such as the offset, scale, and color of the hidden Gaussian points. This enables it to be adapted to the HDGER proposed by SecureGS. For the RDO strategy, our approach is not limited to the growth and pruning of anchor points. In fact, asynchronous gradient accumulation, point cloud clustering for bounding box generation, and adaptive reduction of gradient splitting thresholds can be applied to any GS-based rendering pipeline.
> > > - Although the setup of parameters is important for Scaffold-GS, the parameters related to steganography are relatively few and are not very sensitive. The key parameters related to RDO are actually just **the size of the bounding box** and **decay factor of the gradient threshold**. According to our experiments, we found that these two parameters can generally be set to 2.5 and 4 for different scenes. If needed, we can provide more experimental results regarding the parameter settings.
> > >
> > >
> > >
> > > Thank you very much for your comments. If you have any further questions, feel free to continue the discussion with us.

---

> > > > ### Comment · Reviewer_yt9u · 2024-11-27
> > > >
> > > > Thank you. All my remaining questions have been resolved. I tested the code, and the results are similar to those reported by the authors. While the choice of parameters is not entirely without impact, its sensitivity is significantly reduced compared to Scaffold-GS.
> > > >
> > > > This paper has seen considerable improvement compared to the initial draft. I appreciate the authors’ efforts, and I will raise my score to a 5. However, the reason for not giving a higher score is that, while the paper is indeed interesting, as we discussed regarding W2, the overall progress beyond the enhancements brought by Scaffold-GS is relatively minor. Moreover, the proposed solution for hidden object point clouds does not strike me as sufficiently innovative.
> > > >
> > > > In fact, the most compelling contribution of this paper, in my view, is the implicit neural Gaussian decoding. I appreciate the authors’ reflections on GS-hider in W1 and their proposal of implicit neural Gaussian decoding to address the identified issues. However, this part of the design bears too much resemblance to Scaffold-GS, making it difficult for me to fully consider it an original contribution by the authors.

---

> > > > > ### Author Response · Authors · 2024-11-29
> > > > > **Thanks to Reviewer yt9u for recognition of our work.**
> > > > >
> > > > > Dear Reviewer yt9u:
> > > > >
> > > > > Thank you for your thoughtful follow-up and for acknowledging the improvements in our paper. We appreciate your efforts in testing our code and your decision to raise your score, which really encourages us. We believe that the design philosophy behind SecureGS is different from that of Scaffold-GS. SecureGS focuses more on the unique file format security and geometric structure security inherent to GS steganography, which has not been considered by other methods. Thus, we believe that our work is innovative and has practical values. Nevertheless, we would like to thank you again for your detailed review comments and recognition, which made our work clearer and more complete.
> > > > >
> > > > > Best Regards,
> > > > >
> > > > > Authors of #326

---

### Official Review · Reviewer_HhF8 · 2024-10-31

**Soundness:** 3
**Presentation:** 3
**Contribution:** 3
**Rating:** 6
**Confidence:** 2

**Summary:**

This paper studies a safe and efficient 3D steganography framework. The proposed SecureGS addresses issues existed in previous NeRF-based steganography solutions through a hybrid decoupled Gaussian encryption representation, which secures both file format and geometric structure while concealing 3D objects, images, and bits within anchor points that can only be decoded by authorized users. SecureGS incorporates a region-aware density optimization (RDO) strategy, dynamically adjusting anchor point growth to avoid leaking hidden content.

**Strengths:**

1. SecureGS provides dual security by protecting both the file format and geometric structure, which reduces risks of hidden information exposure in public files.

2. The SecureGS framework achieves real-time rendering rates, nearly three times faster than GS-Hider, with storage requirements that are substantially lower.

3. Evaluation results demonstrate SecureGS's resilience to anchor point pruning, achieving reliable decoding.

**Weaknesses:**

1. Although the author mentioned three shortcomings of existing methods in Introduction, the description is not intuitive and difficult to understand. For example, the concepts of format security and geometric structure security are not explained clearly at the part. I suggest that the current Fig. 2 should be shown at the beginning rather than the pipeline of the proposed method, and the relevant concepts should be described in more detail.

2. The authors claimed that the proposed method addresses the issue of rendering quality existing previous methods. However, the experimental results show that the average improvement is only 1.16 dB. In my experience, such slight changes in PSNR values ​​usually result in only negligible subjective differences. Additionally, the $PSNR_o$ values in Table 2 and 4 are relatively low (below 30 dB), so I think the high-fidelity restoration of hidden objects comes at the expense of the quality of the original scene restoration. This further raises concerns about the fairness of the comparison in Table 1 and 5, because the author did not show the results of $PSNR_0$.

**Questions:**

None

**Details Of Ethics Concerns:**

N.A.

---

> ### Author Response · Authors · 2024-11-21
> **Response to Reviewer HhF8**
>
> Thank you for your constructive feedback! We hope our responses have addressed your concerns. If there are any areas that require further clarification, please don’t hesitate to reach out for further discussion!
>
> > **W1: About some description in Introduction**
>
> Thanks for your reminder. Format security ensures that the published anchor point cloud does not include any additional attributes that might raise suspicion or lead to deletion by malicious attackers. Geometric structure security focuses on guaranteeing that visualizing the Gaussian point cloud does not reveal any trace of the hidden object's geometric structure. **Following your suggestion, we have added this explanation to Line 65-69 of introduction and moved Figure 2 to the beginning to make our paper clearer**.
>
>
>
> > **W2: About rendering quality**
>
> - In the field of 3D rendering, 1.16 dB improvement is already considered a significant performance gain. Figure 5 clearly illustrates the subjective quality improvements of our method compared to other steganography methods. Furthermore, it is worth noting that the rendering quality of the hidden scene has also improved by 1.66 dB compared to GS-Hider, which is another notable and important gain achieved by our method.
> - The PSNR_o in Table 2 represents the fidelity of our method **after undergoing disruptions (e.g., random pruning)**. It reflects the degree of degradation rather than the rendering quality of our method. Table 4 presents our ablation experiments, which are designed to show the advantages of two key modules in our method. The PSNR_o values of the two variants in the table also do not reflect the fidelity of our method.
> - In fact, to the best of our knowledge, no rendering method in the 3DGS field has achieved a PSNR exceeding 30 dB. Our method surpasses all GS-based steganography methods in terms of fidelity and is comparable to the original Scaffold-GS and 3DGS.
> - **We show the PSNR_o result in the "Avg" Column of Table 1 and Table 5**.

---

> > ### Comment · Reviewer_HhF8 · 2024-11-25
> >
> > Thank you for your reply. But I still think that this work's contribution to high fidelity is limited. I decide to maintain the score.

---

> > > ### Author Response · Authors · 2024-11-25
> > > **Thank you for your efforts and positive rating**
> > >
> > > Thank you for your efforts and positive rating. Regarding our contribution to high fidelity, we would like to provide some additional explanations.
> > >
> > > - **Compared to the SOTA 3DGS steganography method, GS-Hider**: SecureGS achieves fidelity improvements of 1.16 dB/1.66 dB for the original scene and hidden information in object hiding tasks (**See Table 1**), and 1.38 dB/4.57 dB in scene hiding tasks (**See Table 7**). These are significant and noteworthy fidelity enhancements, which are attributed to our novel hybrid decoupled Gaussian encryption representation and region-aware density optimization strategies.
> > >
> > > - **Compared to 3DGS and Scaffold-GS**: For the steganography tasks discussed in this paper, Scaffold-GS and 3DGS represent the upper limits of rendering quality for the original scene. **The fidelity of our method has approached, or even slightly exceeded, these theoretical upper bounds**.
> > >
> > > - Lastly, we want to emphasize that **evaluating a 3D steganography algorithm requires a comprehensive consideration of fidelity, security, robustness, and rendering speed**. SecureGS enhances all these aspects, particularly addressing the previously overlooked security concerns, thereby significantly increasing the practicality of the 3DGS steganography framework.
> > >
> > > If needed, we welcome you to continue discussing with us. Please also share the criteria you use to evaluate fidelity contributions. Once again, thanks for your review and positive recognition.

---

### Official Review · Reviewer_6qod · 2024-10-31

**Soundness:** 4
**Presentation:** 4
**Contribution:** 4
**Rating:** 8
**Confidence:** 4

**Summary:**

The paper presents SecureGS, a novel method for 3D Gaussian splatting (3DGS) steganography that addresses significant security and fidelity challenges in protecting 3DGS-based assets. It proposes a hybrid decoupled Gaussian encryption mechanism and a region-aware density optimization strategy, enabling efficient embedding and retrieval of hidden 3D objects, images, and bits within original 3D scenes.

**Strengths:**

(+) Previous steganography lacks a strategy for the explicit point cloud from 3DGS. The paper introduces an interesting and original approach to 3DGS steganography, emphasizing security and fidelity. This innovative framework could pave the way for advancements in the field, inspiring further research into secure 3D content transmission.

(+) Overall, this paper is well-written and has a logical structure. The motivation and contributions are clearly articulated, making it accessible to readers. The figures and diagrams effectively illustrate complex concepts, enhancing reader comprehension.

(+) The authors conduct comprehensive experiments that validate the generalizability and effectiveness of the proposed method across various scenarios. These experiments demonstrate the method's superiority and provide valuable insights into its practical applicability in different contexts.

**Weaknesses:**

(-) The proposed methods, particularly the hybrid encryption and optimization strategies, may introduce complexity that could hinder practical adoption. This complexity might deter practitioners who require straightforward solutions for embedding and retrieving hidden information.

(-) The enhancements in security and fidelity may come at the cost of increased computational requirements, which could limit real-time applications. If the computational demands are too high, they may negate the benefits of improved rendering fidelity in time-sensitive environments.

Minor suggestions:

More vspace between Line 65 and Line 66.

**Questions:**

1.	How does SecureGS compare to traditional steganography methods in terms of computational efficiency and resource requirements?

2.	What specific measures have you taken to ensure the robustness of your method against various types of attacks or degradation, especially in practical deployment scenarios?

3.	Given the complexity of the proposed methods, how do you envision simplifying the implementation for practitioners who may lack advanced technical expertise in 3D steganography?

---

> ### Author Response · Authors · 2024-11-21
> **Response to Reviewer 6qod**
>
> Thank you for your constructive comments! If there are any additional comments to be added, please continue the discussion with us.
>
>
>
> > **W1: Introduce complexity that could hinder practical adoption**
>
> - Our method is essentially an end-to-end black box. Users only need to input the training views corresponding to the original scene and the hidden object to obtain the anchor point cloud with hidden message and the secret MLP. Compared to training the original 3DGS or the steganography method GS-Hider, it does not introduce any additional steps.
> - Our method offers additional functions compared to steganography methods like GS-Hider. For instance, it can directly output the point cloud representing the hidden scene, as well as hide both 3D scenes, bits, or background-free objects. This significantly enhances convenience for practitioners.
>
>
>
> > **W2: Increased computational requirements**
>
> Although our method introduces additional computational overhead compared to Scaffold-GS, it remains resource-efficient compared to all other GS-based steganography methods and the original 3DGS.
>
> - Our average point cloud file memory size is only **267.39 MB**, which is **201.24 MB** less than GS-Hider and **839.28 MB** less than 3DGS+StegaNeRF.
> - Our rendering speed is **131.71 FPS**, with a peak of **144.46 FPS**, which is comparable to the original 3DGS and significantly faster than GS-Hider.
>
>
>
> > **W3: Minor Issues**
>
> Thank you for your reminder. We have corrected the vspace in our paper.
>
>
>
> > **Q1:  Comparison to traditional steganography methods**
>
> Compared to traditional NeRF watermarking, which typically decodes hidden messages from rendered RGB views, our SecureGS offers faster rendering speeds but requires more storage space to store anchor point clouds. This is inherently determined by the characteristics of the 3DGS representation. Additionally, the computational requirements for our SecureGS are similar to those of NeRF watermarking and can be run on a single NVIDIA GTX 3090 GPU.
>
>
>
> > **Q2:  Specific measures taken to ensure the robustness**
>
> Thanks for your valuable comments. Our RDO strategy ensures that even under degradation or attacks, the hidden message can still be decoded effectively by encouraging the splitting of more anchor Gaussian points in key regions of the hidden information. To further enhance the robustness of our method, we may need to incorporate some differentiable degradation operations, such as Gaussian noise addition or random dropout, during the training process. We plan to implement this in our future work.
>
>
>
> > **Q3: Simplifying the implementation for practitioners**
>
> Although our method employs a relatively complex hybrid 3D representation and RDO strategy, these complexities are entirely transparent to users. During the training phase, users only need to input paired training views, and the method outputs Gaussian anchor point clouds and MLPs. During the verification phase, users simply input the anchor point clouds and encrypted MLPs to extract the hidden information.  Furthermore, we will design an interactive editing interface, and provide detailed usage instructions to facilitate practitioners in adopting our method.

---

> > ### Comment · Reviewer_6qod · 2024-11-22
> >
> > Thank you for your response. Your response has addressed my concerns regarding robustness and computational requirements. I hope the authors can further simplify the implementation of 3D steganography in future work to enhance its practicality. To me, this work is insightful. I will maintain my score.

---

> > > ### Author Response · Authors · 2024-11-24
> > > **Thank Reviewer 6qod for recognizing our work**
> > >
> > > Dear Reviewer 6qod:
> > >
> > > Thank you for your prompt response, valuable comments, and recognition of our work. This has helped make our work more complete.
> > >
> > > Best Regards,
> > >
> > > Authors of # 326

---

### Official Review · Reviewer_Keb4 · 2024-11-01

**Soundness:** 3
**Presentation:** 3
**Contribution:** 3
**Rating:** 8
**Confidence:** 5

**Summary:**

The paper proposes SecureGS, a new 3D Gaussian splatting (3DGS) steganography framework that improves the security and fidelity of hiding information in 3D scenes.  Existing methods struggle with fidelity, speed, and security, especially regarding the point cloud's geometric structure.
SecureGS uses a hybrid decoupled Gaussian encryption, concealing hidden Gaussian point positions with a privacy-preserving offset predictor only authorized users can access.  A density region-aware anchor growing strategy further enhances security by dynamically adjusting anchor point density to mask the hidden object's geometry without sacrificing speed or storage.
Experiments show SecureGS significantly outperforms the existing method GS-Hider, especially for geometry security.

**Strengths:**

1. SecureGS proposes a hybrid decoupled Gaussian encryption to embed information within anchor points, accessible only to authorized users.
2. A density region-aware anchor growing and pruning strategy improves security by preventing exposure of the hidden object.
3.  SecureGS achieves this by ensuring both file format and geometric structure security, compared with the existing baselines GS-Hider and 3DGS+StegaNeRF.

**Weaknesses:**

1. While the proposed region-aware density optimization effectively utilizes growing 3D Gaussian points, it results in an increased model size, which is a key concern in 3DGS.
Actions should be taken to implement specific strategies aimed at mitigating the increased model size.
Additionally, the tradeoff between enhanced security and the resulting model size should be thoroughly discussed to provide a clearer perspective on the implications of the proposed approach.

2. The bit accuracy relies on an MLP layer, which is per-scene optimized, which may lead to an unfair comparison setting with the previous baselines CopyRNeRF and NeRFProtector. Why the per-scene optimization-based MLP method is chosen here instead of considering generalizable message extraction methods such as HiDDeN for bit message extraction?

**Questions:**

1. Is the proposed method compatible with the GS compression method such as LightGaussian or CompactGS? What are the potential challenges in integrating compression with the proposed steganography approach?
2. Regarding the 3DGS having a point cloud geometry format, how is the 3D robustness towards 3D distortions? Such as Gaussian noise with a Gaussian Kernel $\sigma$, Dropout operation to randomly remove a fraction of the points, and Cropout operations to remove points in a specific 3D bounding box volume.
3. What if the attacker knows the encryption method and tries to remove or add perturbations on the anchor points in the 3DGS model?

---

> ### Author Response · Authors · 2024-11-21
> **Response to Reviewer Keb4**
>
> Thank you for your constructive comments! If there are any additional comments to be added, please continue the discussion with us.
>
> > **W1: Increased model size**
>
> - Although the use of RDO slightly increased our model's memory usage, it is still **201.24 MB** less than GS-Hider and **839.28 MB** less than 3DGS+StegaNeRF, representing a significant advancement compared to existing 3DGS steganography methods. Moreover, SecureGS achieves up to **144.46 FPS**, maintaining good real-time rendering capabilities.
>
> - Our point cloud file actually includes anchor points for rendering both the original scene and the hidden object (or scene). Therefore, directly comparing our memory size with methods like Scaffold-GS is relatively unfair.
>
> - Our method allows controlling the growth rate of anchor points and reducing the size of the point cloud file by adjusting the bounding box size, the gradient splitting threshold, and the frequency of asynchronous gradient accumulation in RDO. Additionally, it can be combined with Gaussian compression or pruning methods (as shown in Table 2 of the main paper) to further reduce model size.
>
>
>
> > **W2: Generalizable message extraction**
>
> - Although the message extraction methods of NeRFProtector and CopyRNeRF exhibit generalization, their entire watermarking pipeline still requires per-scene optimization for a chosen bit. Therefore, our task setup is consistent with these two methods. Meanwhile, our MLP layer is very lightweight and easy to optimize, without increasing the training time of the original 3D scene. Thus, the comparison between our method and CopyRNeRF as well as NeRFProtector is reasonable.
> - Our SecureGS needs to decode hidden information, including bits, from anchor points. **HiDDeN is a watermarking network trained on RGB images and cannot be applied to the encoding and decoding of point clouds.**
> - We will include the unique generalization advantages of NeRFProtector and CopyRNeRF in message extraction in the main paper. Generalized message extraction will be considered in our future work.
>
>
>
> > **Q1: Compatible with the GS compression method**
>
> - The core of our SecureGS is **using implicit MLPs to hide the attributes of Gaussian points, not limited to specific 3D representation**. Thus, it can adapt to various GS compression methods. For example, we can refer to the significance scoring method in LightGaussian, the vector quantization and opacity regularization used in CompGS [1]  to perform the pruning of anchor points, further reducing the memory size of the point cloud file. Furthermore, since Compact-3DGS [2] also uses MLP to decode the color components, our SecureGS can naturally adapt to it.
>
> - The potential challenges in integrating compression methods with our SecureGS lie in the following aspects. **First**, these methods are often designed for compressing spherical harmonic coefficients, so we need to adapt them to compress the feature embedding $\boldsymbol{f}_v$ in SecureGS. **Second**, we should achieve a balance between security, fidelity, and point cloud storage size via appropriate parameter settings.
>
>
>
> [1] CompGS: Smaller and Faster Gaussian Splatting with Vector Quantization.
>
> [2] Compact 3D Gaussian Representation for Radiance Field.
>
>
>
> > **Q2: 3D robustness towards 3D distortions**
>
> Thanks for your valuable suggestions. Following your suggestion, we apply Gaussian noise with ($\sigma$=0.05, 0.1, 0.15) to the feature attribute of anchor point cloud. The metrics are presented as follows and the visualization results are **in Figure 9 of our appendix**. Furthermore, in **Table 2 of the main paper**, we have already shown the effects of random dropout on Gaussian points. Additionally, in **A.5 of the appendix**, we have shown the robustness of SecureGS to point cloud denoising. The fidelity results under these different degradations consistently demonstrate the strong robustness of our SecureGS.
>
> | Condition     | PSNR_o (dB) | SSIM_o | PSNR_s (dB) | SSIM_s |
> | :------------ | :---------: | :----: | :---------: | :----: |
> | Clean         |    27.93    | 0.822  |    38.21    | 0.986  |
> | $\sigma$=0.05 |    27.87    |  0.82  |    37.52    | 0.984  |
> | $\sigma$=0.1  |    27.74    | 0.817  |    37.04    | 0.983  |
> | $\sigma$=0.15 |    27.55    | 0.857  |    36.18    | 0.981  |
>
>
>
> > **Q3: What if the attacker knows the encryption method?**
>
> - Even if the attacker knows our encryption method, the weights of each secret MLP are unique to each scene, making it extremely difficult for unauthorized users to crack.
> - Our robustness experiments ensure that slight removal or addition of perturbations on the anchor points does not affect the rendering quality of the hidden message. If an attacker attempts to forcibly remove or add a significant portion of the anchor points, the rendering quality of the original scene will also be severely degraded, rendering the 3DGS entirely unusable. This means such destruction is meaningless.

---

> > ### Comment · Reviewer_Keb4 · 2024-11-22
> >
> > Thank you for your response.
> > The feedback about the compatibility with model compression and the robustness towards 3D attacks have addressed my concerns.
> > I also suggest including the discussion about the generalizability of message extraction compared with HiDDeN-based methods. Decoding messages from the point clouds can also become a key feature in this work compared with the previous baselines.
> > All in all, I think the proposed method solves a key problem from the GS-Hider problem to hide the geometrical structure, which can make such steganography more practical.
> > I decide to increase my score. Thank you for your efforts.

---

> > > ### Author Response · Authors · 2024-11-24
> > > **Thank Reviewer Keb4 for recognizing our work**
> > >
> > > Dear Reviewer Keb4:
> > >
> > > Thanks for your response and for recognizing our work. According to your valuable advice, we have included the discussion and comparison with HiDDeN-based methods in **A.2 of our appendix**. Thank you sincerely again!
> > >
> > > Best Regards,
> > >
> > > Authors of #326

---

### Official Review · Reviewer_FjJF · 2024-11-04

**Soundness:** 2
**Presentation:** 2
**Contribution:** 2
**Rating:** 3
**Confidence:** 4

**Summary:**

The paper introduces SecureGS, an innovative framework aimed at enhancing the security and fidelity of 3D Gaussian Splatting (3DGS) steganography. It addresses critical challenges in embedding hidden information within 3D scenes, focusing on copyright protection and privacy. SecureGS employs a hybrid decoupled Gaussian encryption mechanism that allows for the secure embedding and retrieval of 3D objects, images, and bits, while maintaining high rendering quality. Key contributions include a density region-aware anchor growing and pruning strategy that enhances security and experimental results demonstrating significant improvements over existing 3DGS methods in rendering fidelity, speed, and overall security. This framework lays a strong foundation for future advancements in 3D steganography, balancing efficiency and security in digital asset management. However, this paper lacks innovation, and the security verification is not sufficient.

**Strengths:**

SecureGS showcases originality through its hybrid decoupled Gaussian encryption mechanism and a density region-aware anchor strategy. The paper is technically fine and clear, the algorithm seems to scale well, and the results on the different datasets compare very favorably with the different baselines. The experimental results that clearly demonstrate its advantages over existing methods. The clarity of the paper enhances its accessibility, guiding readers through complex concepts with effective visuals and precise descriptions.

**Weaknesses:**

A substantive assessment of the weaknesses of the paper. Focus on constructive and actionable insights on how the work could improve towards its stated goals. Be specific, avoid generic remarks. For example, if you believe the contribution lacks novelty, provide references and an explanation as evidence; if you believe experiments are insufficient, explain why and exactly what is missing, etc.

1: SECUREGS and GS-Hider are notably similar, including their formulas, descriptions, and flowcharts. This resemblance suggests that the research content leans more towards optimizing the memory efficiency of GS-Hider rather than introducing substantial advancements. Consequently, the innovative aspects of this work appear insufficient to establish a distinct contribution to the field of 3DGS security.

2: In terms of robustness verification, this paper fails to address the effectiveness of SecureGS against adversarial attacks and lacks experimental results for hiding scenes within scenes. The framework's ability to withstand pruning does not appear to be robust. Furthermore, the source code for the baseline scheme (GS-Hider) has not been made publicly available, and the paper does not provide detailed descriptions of the relevant parameter settings. As a result, the experimental findings lack credibility and may not be convincing to the reader.

3: In terms of memory optimization, many existing memory optimization strategies use less than 20% of the memory of the original 3DGS, while SecureGS still retains a large number of redundant Gaussian points, which may hinder its ability to achieve real-time rendering in future 3DGS applications. This limitation poses a challenge for practical applications, especially in scenarios that require fast processing and high efficiency.

4: It is better to describe the current research status more systematically. The figures in your paper are a bit blurry. Please consider replacing them with clearer ones.

**Questions:**

Please list up and carefully describe any questions and suggestions for the authors. Think of the things where a response from the author can change your opinion, clarify a confusion or address a limitation. This is important for a productive rebuttal and discussion phase with the authors.

How effective is SecureGS in defending against adversarial attacks? Please show the code and ablation experiment parameters of the paper, especially the details of the comparison with GS-hider.

---

> ### Author Response · Authors · 2024-11-21
> **Response to Reviewer FjJF**
>
> Thank you for your comments. We have addressed your concerns point by point.  Please feel free to continue the discussion with us at any time.
>
> > **W1: Difference between SecureGS and GS-Hider**
>
> We kindly remind you that you might have a **misunderstanding** of our method.
>
> - The innovation of our method **does not lie in optimizing memory efficiency but rather in enhancing the security, fidelity, and rendering speed** of existing 3DGS steganography methods. (**Highlighted in Line 49-69 of our main paper**)
> - Apart from the task setup of 3DGS steganography, our foundational principles, method design, and implemented functions are totally different from GS-Hider.
>   - GS-Hider is based on the original 3DGS theory, while our SecureGS is based on the neural Gaussian theory.
>   - Our SecureGS can decouple the two different groups of Gaussian points for representing the original scene and hidden object, while GS-Hider couples them together via a coupled secured feature attribute.
>   - In addition to the functions provided by GS-Hider, SecureGS can achieve several unique features, including hiding a background-free object, embedding bits, and separating the point cloud of the hidden object from the original scene.
>
> - The few formulas and figures in our paper that are similar to GS-Hider are included either to review prior work or to provide foundational knowledge. These are standard elements shared by all 3DGS papers and **do not diminish our innovation**. For example, Eq. (1) and (2) introduce the rendering principles of 3DGS, and Figure 2 explains the approach of GS-Hider.
>
>
>
> > **W2 and Q1: About robustness verification and adversarial attacks**
>
> - We want to emphasize that robustness in steganography denotes that **hidden message remains well-preserved after degradation**, rather than the fidelity of the original scene remaining unaffected. In Table 2, even after randomly pruning 20% of the points, the fidelity of the hidden object remains above 33 dB. This is further supported by **Figure 10 in the appendix**. Thus, our method demonstrates strong resistance to random pruning degradation.
>
> - To our knowledge, adversarial attacks denote adding perturbations to deceive models, causing them to make incorrect predictions or behave unexpectedly. **This is not closely related to robustness testing in steganography.** We are unsure which specific type of adversarial attack you are referring to; if possible, please clarify it.
>
>
>
> > **W2: About Scene hiding**
>
> Thanks for your advice. We have added the results for hiding scenes within scenes below. Both for the fidelity of the original scene and the hidden scene, our method significantly outperforms GS-Hider. Relevant part is included in **A.4 of our appendix**.
>
> | Method   | PSNR_o (dB) | SSIM_o | LPIPS_o | PSNR_s (dB) | SSIM_s | LPIPS_s |
> | :------- | :---------: | :----: | :-----: | :---------: | :----: | :-----: |
> | GS-Hider |    25.82    | 0.783  |  0.246  |    25.18    | 0.780  |  0.306  |
> | SecureGS |    27.20    | 0.796  |  0.235  |    29.75    | 0.858  |  0.211  |
>
>
>
> > **W2 and Q1: Reproduction of GS-Hider**
>
> We requested the official code from the authors of GS-Hider to ensure that our comparisons are fair and valid. Our code will be made publicly available after the review process is completed. Specifically, we reproduce GS-Hider via its coupled feature rasterizer, dividing the screen into 16×16 tiles. The dimension of feature attribute is set to 16. Two decoders are composed of 5  Conv layers. All the parameters use the default settings in GS-Hider.
>
>
>
> > **W3: About memory optimization and real-time rendering**
>
> - Since our SecureGS needs to render both the original scene and the hidden scene (objects), it is **unfair** to directly compare SecureGS with some GS pruning methods that only render the original scene.
>
> - Our average storage size is 267.39 MB, compared to 796.406 MB for 3DGS. With the 20% random point pruning from Table 2, our memory usage is only **26\%** of the original 3DGS, making it highly lightweight.
>
> - We have reported our FPS in Tables 1 and 5, where it reaches up to **131.71 FPS** in scene-level and **144.46 FPS** in single-image hiding, far exceeding the real-time requirements (30 FPS). Our rendering time is comparable to Scaffold-GS and 3DGS.
>
> - We kindly remind you again, that although our method shows good memory efficiency, the main innovation of our approach does not lie in optimizing memory usage. Therefore, these memory optimization methods are **orthogonal** to our work.
>
>
>
> > **W4: Current research status and unclear figures**
>
> In Section 2 and Section 3.1, we have systematically introduced the advancements in 3DGS and 3D steganography, especially analyzing the issues in the 3DGS-based work, GS-Hider. If there are any unclear aspects, please feel free to point out the specific areas, including any blurry figures.

---

> ### Author Response · Authors · 2024-11-29
>
> Dear Reviewer FjJF,
>
> Thanks a lot for your previous constructive comments. We would like to know if our revisions have addressed your concerns? If required, please do not hesitate to continue the discussion with us.
>
> Best Regards,
>
> Authors of #326.

---

### Author Response · Authors · 2024-11-21
**Global Response**

We sincerely thank all five reviewers for their constructive comments. All reviewers have acknowledged our innovations, methodological contributions, and experimental results. We have responded to the reviewers' concerns point by point and revised our main paper accordingly. The revised sections are highlighted in $\textcolor{red}{\text{red}}$. We humbly expect you can check the **replies & revised paper** and reconsider the decision.

---

### Author Response · Authors · 2024-11-25
**Looking Forward to Further Discussions**

Dear reviewers:

We appreciate the time you dedicated to reviewing our work and your recognition of our work. Regarding the concerns you raised, we have provided explanations in our responses. We would like to ensure that your concerns have been adequately addressed. If there are any aspects of our work that remain unclear to you, please don't hesitate to let us know.

Best regards,

Authors of #326

---

### Meta-Review · Area_Chair_BkCS · 2024-12-19

**Metareview:**

SecureGS effectively leverages the strengths of Scaffold-GS and GS-Hider to achieve better fidelity and security in information hiding.
Its region-aware anchor growth and pruning strategy are designed to prevent hidden information from being exposed by adaptively optimizing anchor density. The implicit Gaussian neural decoding is also innovative suggested by the reviewers. Overall, this paper received positive feedback from the reviewers, and AC agrees with the recommendations of the reviewers.

**Additional Comments On Reviewer Discussion:**

Most of the reviewers have been convinced by the authors' rebuttal. Even though reviewer yt9u thought the contribution brought by Scafforld-GS might not be significant, the reviewer overall satisfied with the authors' rebuttal and raised the score.
Reviewer FjjF gave a negative rate and provided comments on the weakness of this work. In the rebuttal period, the reviewer did not confirm whether the comments have addressed the concerns. AC thus read the rebuttal and thinks most of the comments have been well addressed by the authors.

---

### Decision · Program_Chairs · 2025-01-22

Accept (Poster)